# A Theory of Unimodal Bias in Multimodal Learning

## Abstract

Using multiple input streams simultaneously in training multimodal neural networks is intuitively advantageous, but practically challenging. A key challenge is unimodal bias, where a network overly relies on one modality and ignores others during joint training. While unimodal bias is well-documented empirically, our theoretical understanding of how architecture and data statistics influence this bias remains incomplete. Here we develop a theory of unimodal bias with deep multimodal linear networks. We calculate the duration of the unimodal phase in learning as a function of the depth at which modalities are fused within the network, dataset statistics, and initialization. We find that the deeper the layer at which fusion occurs, the longer the unimodal phase. In addition, our theory reveals the modality learned first is not necessarily the modality that contributes more to the output. Our results, derived for multimodal linear networks, extend to ReLU networks in certain settings. Taken together, this work illuminates pathologies of multimodal learning under joint training, showing that late and intermediate fusion architectures can give rise to long unimodal phases and even prioritize learning a less helpful modality.

## 1 Introduction

The success of multimodal deep learning hinges on effectively utilizing multiple modalities (Baltrušaitis et al., 2019; Liang et al., 2022). However, some multimodal networks overly rely on a faster-to-learn or easier-to-learn modality and ignore the others during joint training (Goyal et al., 2017; Cadene et al., 2019; Wang et al., 2020; Gat et al., 2021; Peng et al., 2022). For example, Visual Question Answering models should provide a correct answer by both "listening" to the question and "looking" at the image (Agrawal et al., 2016), whereas they tend to overly rely on the language modality and ignore the visual modality (Goyal et al., 2017; Agrawal et al., 2018; Hessel & Lee, 2020). This phenomenon has been observed in a variety of settings, and has several names: unimodal bias (Cadene et al., 2019), greedy learning (Wu et al., 2022), modality competition (Huang et al., 2022), modality laziness (Du et al., 2023), and modality underutilization (Makino et al., 2023).

In this paper we adopt the term unimodal bias to refer to the phenomenon in which a multimodal network learns from different input modalities at different times during joint training. The extent to which multimodal networks exhibit unimodal bias depends on both the dataset and the multimodal network. Goyal et al. (2017); Agrawal et al. (2018); Hudson & Manning (2019) tried to alleviate the bias by building more balanced multimodal datasets. Empirical work has shown that unimodal bias emerges in jointly trained late fusion networks (Wang et al., 2020; Huang et al., 2022) and intermediate fusion networks (Wu et al., 2022), while early fusion networks may encourage usage of all input modalities (Gadzicki et al., 2020; Barnum et al., 2020).

Despite empirical evidence, there is scarce theoretical understanding of how unimodal bias arises and how it is affected by the network configuration, dataset statistics, and initialization. To unravel this pathological behavior, we study deep multimodal linear networks with three common fusion schemes: early, intermediate, and late (Ramachandram & Taylor, 2017). We find that unimodal bias is absent in early fusion linear networks but present in intermediate and late fusion linear networks. Intermediate and late fusion linear networks learn one modality first and the other after a delay, yielding a phase in which the multimodal network implements a unimodal function. This difference in time is consistent with the empirical observation that multimodal networks learn different

modalities at different speeds (Wang et al., 2020; Wu et al., 2022). We compute the duration of the unimodal phase in terms of parameters of the network and the dataset. We find that a deeper fusion layer within the multimodal network, stronger correlations between input modalities, and greater disparities in input-output correlations for each modality all prolong the unimodal phase. We also find that multimodal networks have a superficial preference for which modality to learn first: they prioritize the faster-to-learn modality, which is not necessarily the modality that contributes more to the output, and we derive conditions under which a network will exhibit this superficial preference. Our results apply to ReLU networks in certain settings, providing insights for examining, diagnosing, and curing unimodal bias in a broader range of realistic cases.

Our contributions are the following: (i) We provide a theoretical explanation for the presence of unimodal bias in late and intermediate fusion linear networks and the absence of unimodal bias in early fusion linear networks. (ii) We calculate the duration of the unimodal phase in mulitmodal learning with late and intermediate fusion linear networks, as a function of the network configuration, correlation matrices of the dataset, and initialization scale. (iii) We show that under specific conditions on dataset statistics, late and intermediate fusion linear networks exhibit a superficial modality preference, prioritizing learning a modality that makes less contribution to the output. (iv) We validate our findings with numerical simulations of deep linear networks and certain two-layer ReLU networks.

## 1.1 RELATED WORK

Several attempts have been made to explain the unimodal bias behavior. Various metrics have been proposed to inspect the internal mechanics of multimodal learning and multimodal models (Wang et al., 2020; Wu et al., 2022; Kleinman et al., 2023). Huang et al. (2021; 2022) provides a theoretical explanation for why multimodal learning has the capacity to outperform unimodal learning but can fail to deliver. Makino et al. (2023) propose incidental correlation to diagnose and explain modality underutilization in the small data regime. Our work differs by seeking an analytical relationship between unimodal bias, network configuration, and dataset statistics.

Our work leverages a rich line of theoretical literature on deep linear neural networks. Exact solutions and reductions of the gradient descent dynamics have been derived in (Fukumizu, 1998; Saxe et al., 2014; 2019; Arora et al., 2018; Advani et al., 2020; Atanasov et al., 2022; Shi et al., 2022). Balancing properties in linear networks were discovered and proved in (Ji & Telgarsky, 2019; Du et al., 2018). However, as multimodal linear networks are not fully connected and multimodal datasets generally do not have white input covariance, previous solutions no longer apply, and we had to develop new analytic tools.

## 2 PROBLEM SETUP

### 2.1 MULTIMODAL DATA

Let $\boldsymbol{x} \in \mathbb{R}^D$ represent an arbitrary multimodal input and $y \in \mathbb{R}$ be its scalar target output. We are given a dataset $\{\boldsymbol{x}^\mu, y^\mu\}_{\mu=1}^P$ consisting of $P$ samples. For simplicity, we assume there are two modalities A and B with full input $\boldsymbol{x} = [\boldsymbol{x}_\text{A}, \boldsymbol{x}_\text{B}]^\top$. Since we study multimodal linear networks, the learning dynamics only depend on correlation matrices of the dataset (Fukumizu, 1998; Saxe et al., 2014). We notate the input correlation matrix as $\boldsymbol{\Sigma}$ and input-output correlation matrix as $\boldsymbol{\Sigma}_{y\boldsymbol{x}}$ defined as

$$\boldsymbol{\Sigma} = \begin{bmatrix} \boldsymbol{\Sigma}_\text{A} & \boldsymbol{\Sigma}_\text{AB} \\ \boldsymbol{\Sigma}_\text{BA} & \boldsymbol{\Sigma}_\text{B} \end{bmatrix} = \begin{bmatrix} \langle \boldsymbol{x}_\text{A} \boldsymbol{x}_\text{A}^\top \rangle & \langle \boldsymbol{x}_\text{A} \boldsymbol{x}_\text{B}^\top \rangle \\ \langle \boldsymbol{x}_\text{B} \boldsymbol{x}_\text{A}^\top \rangle & \langle \boldsymbol{x}_\text{B} \boldsymbol{x}_\text{B}^\top \rangle \end{bmatrix}, \boldsymbol{\Sigma}_{y\boldsymbol{x}} = \begin{bmatrix} \boldsymbol{\Sigma}_{y\boldsymbol{x}_\text{A}} & \boldsymbol{\Sigma}_{y\boldsymbol{x}_\text{B}} \end{bmatrix} = \begin{bmatrix} \langle y\boldsymbol{x}_\text{A}^\top \rangle & \langle y\boldsymbol{x}_\text{B}^\top \rangle \end{bmatrix} \quad (1)$$

where $\langle \cdot \rangle$ denotes the average over the dataset. We assume data points are centered to have zero mean $\langle \boldsymbol{x} \rangle = \boldsymbol{0}$ and covariance $\boldsymbol{\Sigma}$ has full rank, but make no further assumptions on the correlation matrices.

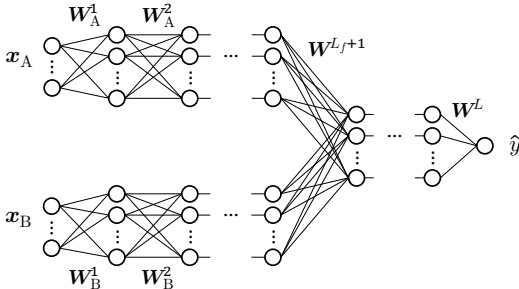

Figure 1: Schematic of a multimodal fusion network with total depth $L$ and fusion layer at $L_f$.

## 2.2 MULTIMODAL FUSION LINEAR NETWORK

We study a deep linear network with total depth $L$ and fusion layer at $L_f$ defined as

$$\hat{y}(\boldsymbol{x}; \boldsymbol{W}) = \prod_{j=L_f+1}^{L} \boldsymbol{W}^j \left( \prod_{i=1}^{L_f} \boldsymbol{W}_{\mathrm{A}}^i \boldsymbol{x}_{\mathrm{A}} + \prod_{i=1}^{L_f} \boldsymbol{W}_{\mathrm{B}}^i \boldsymbol{x}_{\mathrm{B}} \right) \equiv \boldsymbol{W}_{\mathrm{A}}^{\mathrm{tot}} \boldsymbol{x}_{\mathrm{A}} + \boldsymbol{W}_{\mathrm{B}}^{\mathrm{tot}} \boldsymbol{x}_{\mathrm{B}} \equiv \boldsymbol{W}^{\mathrm{tot}} \boldsymbol{x}. \quad (2)$$

The overall network input-output map is denoted as $\boldsymbol{W}^{\mathrm{tot}}$ and the map for each modality is denoted as $\boldsymbol{W}_{\mathrm{A}}^{\mathrm{tot}}, \boldsymbol{W}_{\mathrm{B}}^{\mathrm{tot}}$. We use $\boldsymbol{W}$ to denote all weight parameters collectively. We assume the number of neurons in both pre-fusion layer branches is of the same order. A schematic of this network is given in Fig. 1.

Our network definition incorporates bimodal deep linear networks of three common fusion schemes. Categorized by the multimodal deep learning community (Ramachandram & Taylor, 2017), the case where $L_f = 1$ is early or data-level fusion; $1 < L_f < L$ is intermediate fusion; and $L_f = L$ is late or decision-level fusion.

## 2.3 GRADIENT DESCENT DYNAMICS

The network is trained by optimizing the mean square error loss $\mathcal{L} = \frac{1}{2P} \sum_{\mu=1}^{P} (\hat{y}^\mu - y^\mu)^2$ with full batch gradient descent. In the limit of small learning rate, the gradient descent dynamics are well approximated by the continuous time differential equations given below; see Appendix A. For pre-fusion layers $1 \leq l \leq L_f$,

$$\tau \dot{\boldsymbol{W}}_{\mathrm{A}}^l = \left( \prod_{j=L_f+1}^{L} \boldsymbol{W}^j \prod_{i=l+1}^{L_f} \boldsymbol{W}_{\mathrm{A}}^i \right)^\top \left( \boldsymbol{\Sigma}_{y\boldsymbol{x}_{\mathrm{A}}} - \boldsymbol{W}_{\mathrm{A}}^{\mathrm{tot}} \boldsymbol{\Sigma}_{\mathrm{A}} - \boldsymbol{W}_{\mathrm{B}}^{\mathrm{tot}} \boldsymbol{\Sigma}_{\mathrm{BA}} \right) \left( \prod_{i=1}^{l-1} \boldsymbol{W}_{\mathrm{A}}^i \right)^\top, \quad (3a)$$

$$\tau \dot{\boldsymbol{W}}_{\mathrm{B}}^l = \left( \prod_{j=L_f+1}^{L} \boldsymbol{W}^j \prod_{i=l+1}^{L_f} \boldsymbol{W}_{\mathrm{B}}^i \right)^\top \left( \boldsymbol{\Sigma}_{y\boldsymbol{x}_{\mathrm{B}}} - \boldsymbol{W}_{\mathrm{A}}^{\mathrm{tot}} \boldsymbol{\Sigma}_{\mathrm{AB}} - \boldsymbol{W}_{\mathrm{B}}^{\mathrm{tot}} \boldsymbol{\Sigma}_{\mathrm{B}} \right) \left( \prod_{i=1}^{l-1} \boldsymbol{W}_{\mathrm{B}}^i \right)^\top. \quad (3b)$$

For post-fusion layers $L_f + 1 \leq l \leq L$,

$$\begin{aligned}
\tau \dot{\boldsymbol{W}}^l = {} & \left( \prod_{j=l+1}^{L} \boldsymbol{W}^j \right)^\top \left( \boldsymbol{\Sigma}_{y\boldsymbol{x}_{\mathrm{A}}} - \boldsymbol{W}_{\mathrm{A}}^{\mathrm{tot}} \boldsymbol{\Sigma}_{\mathrm{A}} - \boldsymbol{W}_{\mathrm{B}}^{\mathrm{tot}} \boldsymbol{\Sigma}_{\mathrm{BA}} \right) \left( \prod_{j=L_f+1}^{l-1} \boldsymbol{W}^j \prod_{i=1}^{L_f} \boldsymbol{W}_{\mathrm{A}}^i \right)^\top \\
& + \left( \prod_{j=l+1}^{L} \boldsymbol{W}^j \right)^\top \left( \boldsymbol{\Sigma}_{y\boldsymbol{x}_{\mathrm{B}}} - \boldsymbol{W}_{\mathrm{A}}^{\mathrm{tot}} \boldsymbol{\Sigma}_{\mathrm{AB}} - \boldsymbol{W}_{\mathrm{B}}^{\mathrm{tot}} \boldsymbol{\Sigma}_{\mathrm{B}} \right) \left( \prod_{j=L_f+1}^{l-1} \boldsymbol{W}^j \prod_{i=1}^{L_f} \boldsymbol{W}_{\mathrm{B}}^i \right)^\top, \quad (4)
\end{aligned}$$

where the time constant $\tau$ is the inverse of learning rate $\eta^{-1} = \tau$. We abuse the notation $\prod_j \boldsymbol{W}^j$ to represent the ordered product of matrices with the largest index on the left and smallest on the right. The network is initialized with small random weights.

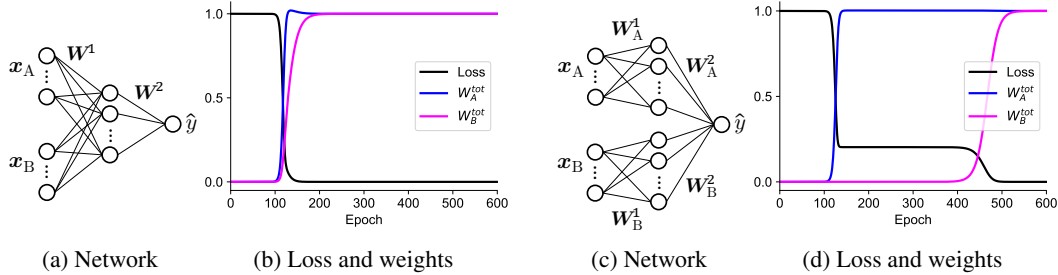

(a) Network      (b) Loss and weights      (c) Network      (d) Loss and weights

Figure 2: Schematic and training trajectories of two-layer early and late fusion linear networks. (a,c) Schematic of a two-layer early fusion (a) and late fusion (c) linear network. (b,d) Loss and total weights trajectories in the two-layer early fusion (b) and late fusion (d) linear network. Details: Both networks are trained to learn the same task of fitting $y = x_A + x_B$. Inputs $x_A$ and $x_B$ are scalars with covariance matrix $\Sigma = \text{diag}(4, 1)$. The early fusion network has 100 hidden neurons. The late fusion network has 100 hidden neurons in both branches. Both networks are initialized with small random weights sampled independently from $\mathcal{N}(0, 10^{-9})$. The learning rate is $0.04$.

## 3    Two-Layer Multimodal Linear Networks

We first study two-layer multimodal linear networks with $L = 2$. There are two possible fusion schemes for two-layer networks according to our setup, namely early fusion $L_f = 1$, as in Fig. 2a and late fusion $L_f = 2$, as in Fig. 2c. Unimodal bias is absent in early fusion linear networks but present in late fusion linear networks. We first give an overview of the qualitative behavior based on the loss landscape and then quantitatively characterize the unimodal bias.

### 3.1    Two-Layer Early Fusion Linear Network

Early fusion networks learn from both modalities simultaneously as shown in Fig. 2b. There is no conspicuous unimodal phase over the course of training.

This behavior can be appreciated through an analysis of the fixed points of the gradient dynamics. There are two manifolds of fixed points in early fusion networks (Appendix B): one is an unstable fixed point at zero $\mathcal{M}_0$ and the other is a manifold of stable fixed points at $\mathcal{M}_*$,

$$\mathcal{M}_0 = \{\boldsymbol{W} | \boldsymbol{W} = \boldsymbol{0}\}; \tag{5a}$$

$$\mathcal{M}_* = \left\{\boldsymbol{W} | \boldsymbol{W}^{\text{tot}} = \boldsymbol{\Sigma}_{y\boldsymbol{x}} \boldsymbol{\Sigma}^{-1}\right\}. \tag{5b}$$

The network starts from small initialization, which is close to the unstable fixed point $\mathcal{M}_0$. When learning progresses, the network escapes from the unstable fixed point $\mathcal{M}_0$ and converges to the global pseudo-inverse solution at $\mathcal{M}_*$. As studied by Saxe et al. (2019); Jacot et al. (2021); Pesme & Flammarion (2023), linear networks trained from small initialization exhibit quasi-stage-like behavior: learning progresses slowly for most of the time and rapidly moves from one fixed point or saddle to the next with a brief sigmoidal transition. Because there is only one fixed point aside from the zero fixed point at initialization to transit to, all dimensions and thus all modalities are learned during this one transition. Since the transitional period is very brief compared to the total learning time, all modalities are learned at almost the same time in early fusion networks.

### 3.2    Two-Layer Late Fusion Linear Network

Late fusion networks learn from two modalities with two sigmoidal transitions at two different times, as shown in Fig. 2d. There can be a prolonged unimodal phase during joint training.

Late fusion linear networks have the same two manifolds of fixed points $\mathcal{M}_0, \mathcal{M}_*$ as early fusion networks. In addition, late fusion linear networks have two manifolds of saddles $\mathcal{M}_A, \mathcal{M}_B$ (Appendix C.1), corresponding to learning one modality but not the other,

$$\mathcal{M}_A = \left\{\boldsymbol{W} | \boldsymbol{W}_A^{\text{tot}} = \boldsymbol{\Sigma}_{y\boldsymbol{x}_A} \boldsymbol{\Sigma}_A^{-1}, \boldsymbol{W}_B^{\text{tot}} = \boldsymbol{0}\right\}; \tag{6a}$$

$$\mathcal{M}_B = \left\{\boldsymbol{W} | \boldsymbol{W}_A^{\text{tot}} = \boldsymbol{0}, \boldsymbol{W}_B^{\text{tot}} = \boldsymbol{\Sigma}_{y\boldsymbol{x}_B} \boldsymbol{\Sigma}_B^{-1}\right\}. \tag{6b}$$

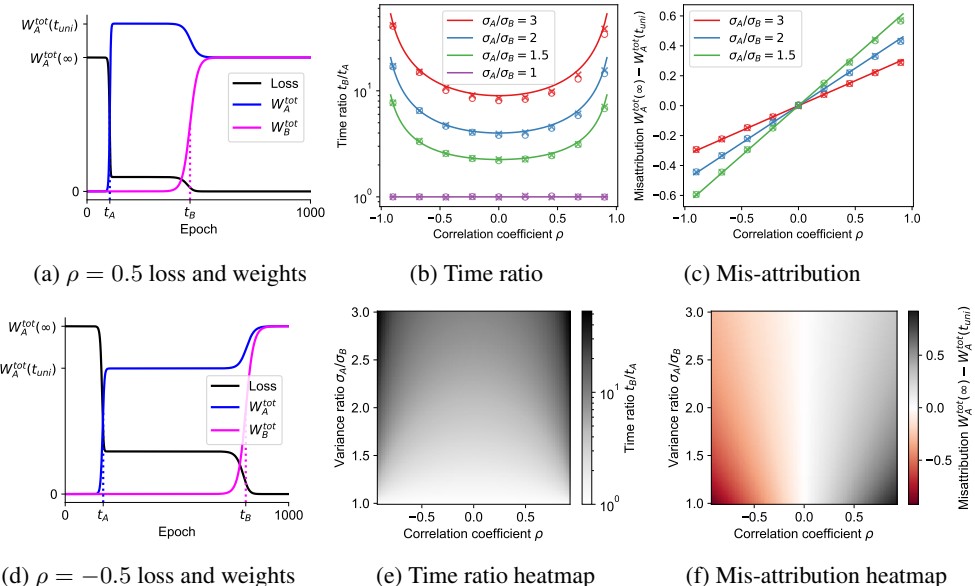

(a) $\rho = 0.5$ loss and weights     (b) Time ratio     (c) Mis-attribution

(d) $\rho = -0.5$ loss and weights     (e) Time ratio heatmap     (f) Mis-attribution heatmap

Figure 3: The duration of the unimodal phase and the amount of mis-attribution in two-layer late fusion linear networks. Two-layer late fusion linear networks of the same setting as Fig. 2c are trained to fit $y = \boldsymbol{x}_A + \boldsymbol{x}_B$ with different input covariance matrices. We notate the elements in the two-dimentional input covariance matrix as $\boldsymbol{\Sigma} = [\sigma_A^2, \rho\sigma_A\sigma_B; \rho\sigma_A\sigma_B, \sigma_B^2]$. (a,d) Examples of loss and total weight trajectories in two-layer late fusion networks when modalities have positive correlation $\rho = 0.5$ (a) and negative correlation $\rho = -0.5$ (d). (b,c,e,f) Time ratio and amount of mis-attribution. Lines and heatmaps are theoretical predictions; circles are simulations of two-layer late fusion linear networks; crosses are simulations of two-layer late fusion ReLU networks. When assuming inputs $\boldsymbol{x}_A, \boldsymbol{x}_B$ are scalars and the target output is $y = \boldsymbol{x}_A + \boldsymbol{x}_B$, the time ratio reduces to $1 + (\frac{\sigma_A^2}{\sigma_B^2} - 1)/(1 - \rho^2)$; the amount of mis-attribution reduces to $\rho\frac{\sigma_B}{\sigma_A}$.

The late fusion linear network therefore goes through two transitions because the network first arrives and lingers near a saddle in $\mathcal{M}_A$ or $\mathcal{M}_B$ and subsequently converges to the global pseudo-inverse solution in $\mathcal{M}_*$. Visiting the saddle gives rise to the plateau in the loss that separates the time when the two modalities are learned. We visualize the learning path and fixed points in the phase portrait in Fig. 6.

Which modality is learned first depends solely on the relative size of $\|\boldsymbol{\Sigma}_{y\boldsymbol{x}_A}\|$ and $\|\boldsymbol{\Sigma}_{y\boldsymbol{x}_B}\|$ in late fusion linear networks with small initialization[1]. For definiteness, in all our simulations we choose $\|\boldsymbol{\Sigma}_{y\boldsymbol{x}_A}\| > \|\boldsymbol{\Sigma}_{y\boldsymbol{x}_B}\|$ which ensures that modality A is learned first; see Appendix C.3.1. We define $t_A$ to be the time when the total weight of modality A reaches half of its associated plateau; and similarly for $t_B$ as illustrated in Figs. 3a and 3d. At time $t_A$, $\boldsymbol{W}_A^{\text{tot}}$ escapes from the zero fixed point $\mathcal{M}_0$ and the network visits a saddle in $\mathcal{M}_A$ where modality A fits the output as much as it can while modality B has not been learned. During the time lag from $t_A$ to $t_B$, the loss lingers in a plateau and the network is unimodal. At time $t_B$, modality B escapes from the zero fixed point and the network converges to the global pseudo-inverse solution fixed point in $\mathcal{M}_*$. We quantify how long the unimodal phase is in Section 3.2.1. We take a closer look at the unimodal mode during the unimodal phase by highlighting the mis-attribution in unimodal phase in Section 3.2.2 and the superficial modality preference in Section 3.2.3.

### 3.2.1 DURATION OF THE UNIMODAL PHASE

We first quantify the duration of the unimodal phase. Because of the small random initialization, we can assume that the Frobenius norm of $\boldsymbol{W}_A^1$ is approximately equal to the norm of $\boldsymbol{W}_B^1$ at initialization, notated as $u_0$. Through leading order approximation, we compute the times $t_A$ and $t_B$ in

---

[1] We use $\|\cdot\|$ to notate the L2 norm of a vector or the Frobenius norm of a matrix in this paper.

Appendix C.3, which yields

$$t_{\mathrm{A}} \approx \tau \|\mathbf{\Sigma}_{y\boldsymbol{x}_{\mathrm{A}}}\|^{-1} \ln \frac{1}{u_0}, \quad t_{\mathrm{B}} \approx t_{\mathrm{A}} + \tau \frac{1 - \|\mathbf{\Sigma}_{y\boldsymbol{x}_{\mathrm{A}}}\|^{-1} \|\mathbf{\Sigma}_{y\boldsymbol{x}_{\mathrm{B}}}\|}{\left\|\mathbf{\Sigma}_{y\boldsymbol{x}_{\mathrm{B}}} - \mathbf{\Sigma}_{y\boldsymbol{x}_{\mathrm{A}}} \mathbf{\Sigma}_{\mathrm{A}}^{-1} \mathbf{\Sigma}_{\mathrm{AB}}\right\|} \ln \frac{1}{u_0}. \tag{7}$$

To compare the unimodal phase duration across different settings, we focus on the time ratio,

$$\frac{t_{\mathrm{B}}}{t_{\mathrm{A}}} = 1 + \frac{\|\mathbf{\Sigma}_{y\boldsymbol{x}_{\mathrm{A}}}\| - \|\mathbf{\Sigma}_{y\boldsymbol{x}_{\mathrm{B}}}\|}{\left\|\mathbf{\Sigma}_{y\boldsymbol{x}_{\mathrm{B}}} - \mathbf{\Sigma}_{y\boldsymbol{x}_{\mathrm{A}}} \mathbf{\Sigma}_{\mathrm{A}}^{-1} \mathbf{\Sigma}_{\mathrm{AB}}\right\|}. \tag{8}$$

We note that the time lag $t_{\mathrm{B}} - t_{\mathrm{A}}$ is proportional to $\|\mathbf{\Sigma}_{y\boldsymbol{x}_{\mathrm{A}}}\| - \|\mathbf{\Sigma}_{y\boldsymbol{x}_{\mathrm{B}}}\|$, which accords with the intuition that the $\|\mathbf{\Sigma}_{y\boldsymbol{x}_{\mathrm{A}}}\|$ governs the speed at which modality A is learned and $\|\mathbf{\Sigma}_{y\boldsymbol{x}_{\mathrm{B}}}\|$ governs the speed at which modality B is learned during time 0 to $t_{\mathrm{A}}$. The denominator governs the speed at which modality B is learned during the unimodal phase from time $t_{\mathrm{A}}$ to $t_{\mathrm{B}}$. Specifically, the network visits a saddle in $\mathcal{M}_{\mathrm{A}}$ during the unimodal phase, which affects the speed at which modality B is learned during the unimodal phase if cross correlation is not zero. Near the saddle, the speed of modality B is reduced by $\mathbf{\Sigma}_{y\boldsymbol{x}_{\mathrm{A}}} \mathbf{\Sigma}_{\mathrm{A}}^{-1} \mathbf{\Sigma}_{\mathrm{AB}}$; see Appendix C.3.

We validate Eq. (8) with numerical simulations in Fig. 3b. From Eq. (8) and Fig. 3b, we conclude that stronger correlations between input modalities and a greater disparity in input-output correlations for each modality make the time ratio larger, indicating a longer unimodal phase. In the extreme case of having maximum correlation, $\boldsymbol{x}_{\mathrm{A}}$ and $\boldsymbol{x}_{\mathrm{B}}$ have collinearity and one of them is redundant. Thus the denominator $\mathbf{\Sigma}_{y\boldsymbol{x}_{\mathrm{B}}} - \mathbf{\Sigma}_{y\boldsymbol{x}_{\mathrm{A}}} \mathbf{\Sigma}_{\mathrm{A}}^{-1} \mathbf{\Sigma}_{\mathrm{AB}} = \mathbf{0}$ and the time ratio is $\infty$, implying later becomes never — the network learns to fit the output only with modality A and modality B will never be learned as shown in Fig. 7.

We also simulate two-layer late fusion networks with ReLU nonlinearity added to the hidden layer, while holding the rest of our setting fixed. We empirically find that two-layer late fusion ReLU networks trained to learn a linear target map learn two modalities with two transitions and the time ratio plotted in Fig. 3b (crosses) closely aligns with the theoretical predictions derived for late fusion linear networks.

### 3.2.2 MIS-ATTRIBUTION IN THE UNIMODAL PHASE

We now quantify how much the multimodal network mis-attributes some of the output to modality A during the unimodal phase. When modalities are correlated, the local pseudo-inverse solution differs from the global pseudo-inverse solution ($\left[\mathbf{\Sigma}_{y\boldsymbol{x}_{\mathrm{A}}} \mathbf{\Sigma}_{\mathrm{A}}^{-1}, \mathbf{\Sigma}_{y\boldsymbol{x}_{\mathrm{B}}} \mathbf{\Sigma}_{\mathrm{B}}^{-1}\right] \neq \mathbf{\Sigma}_{y\boldsymbol{x}} \mathbf{\Sigma}^{-1}$). During the unimodal phase, $\boldsymbol{W}_{\mathrm{A}}^{\mathrm{tot}}$ fits the output as much as it can and the network mis-attributes some of the output contributed by modality B to modality A by exploiting their correlations. Specifically, the weights of modality A overshoot if modalities have a positive correlation as in Fig. 3a and undershoot if negative as in Fig. 3d. This mis-attribution is then corrected when modality B catches up and the network eventually converges to the global pseudo-inverse solution. We demonstrate, using scalar input for clarity, that mis-attribution is more severe when modalities have higher correlation; see Figs. 3c and 3f.

When modalities are uncorrelated, late fusion networks do not mis-attribute during the unimodal phase. Weights for modality A converge to the global pseudo-inverse solution at time $t_{\mathrm{A}}$ and do not change thereafter, as in Fig. 2d, since the local pseudo-inverse solutions are the same as the global pseudo-inverse solution. In this case, late fusion networks behave the same as separately trained unimodal networks.

We empirically find that mis-attribution exists in two-layer late fusion ReLU networks as well and the amount of mis-attribution plotted in Fig. 3c (crosses) closely align with theoretical predictions for late fusion linear networks.

### 3.2.3 SUPERFICIAL MODALITY PREFERENCE

We now look into which modality is learned first. Late fusion networks have what we call "superficial modality preference" for which modality to learn first. They prioritize the modality that is faster to learn, which is not necessarily the modality that yields the larger decrease in loss. Put differently, late fusion networks first learn the modality that has a higher correlation with the output,

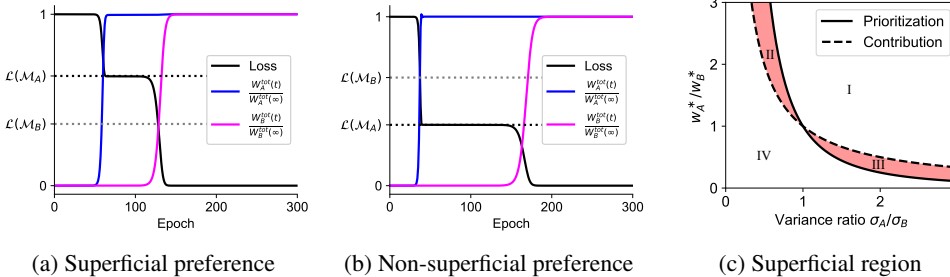

(a) Superficial preference     (b) Non-superficial preference     (c) Superficial region

Figure 4: Demonstration of superficial modality preference. Two-layer late fusion linear networks with the same setting as Fig. 2 are trained with different datasets. (a,b) In both examples, modality A is learned first. The dotted black line marks the loss when the network visits $\mathcal{M}_A$. The dotted gray line marks the loss if the network had instead visited $\mathcal{M}_B$. (a) The target output is $y = x_A + 4x_B$ and the input covariance matrix is $\Sigma = \mathrm{diag}(9, 1)$. The dotted gray line is below the dotted black, meaning modality B contributes more to the output. The prioritized modality is therefore not the modality that contributes more to the output. (b) The target output is $y = x_A + 3x_B$ and the input covariance matrix is $\Sigma = \mathrm{diag}(16, 1)$. The dotted black line is below the dotted gray, meaning modality A contributes more to the output. The prioritized modality is the modality that contributes more to the output. (c) Boundaries of which modality is prioritized and which modality contributes more to the output in terms of dataset statistics. In region I and III, modality A is learned first. In region I and II, modality A contributes more to the output. Thus in region II and III (shaded red), prioritization and contribution disagree, resulting in superficial modality preference.

even though the other modality may make a larger contribution to the output. Under the following two conditions on dataset statistics,

$$\|\Sigma_{yx_A}\| > \|\Sigma_{yx_B}\|, \quad \Sigma_{yx_A}\Sigma_A^{-1}\Sigma_{yx_A}^\top < \Sigma_{yx_B}\Sigma_B^{-1}\Sigma_{yx_B}^\top, \tag{9}$$

modality A is faster to learn but modality B contributes more to the output. We present an example where these conditions hold in Fig. 4a and where they do not in Fig. 4b.

If $x_A, x_B$ are scalars and uncorrelated, the two inequality conditions in Eq. (9) reduce to

$$\frac{\sigma_B^2}{\sigma_A^2} < \frac{w_A^*}{w_B^*} < \frac{\sigma_B}{\sigma_A}, \tag{10}$$

where $\sigma_A, \sigma_B$ are variances of $x_A, x_B$ and we assume the target output is generated as $y = w_A^* x_A + w_B^* x_B$. We plot the two conditions in Fig. 4c. Region III satisfies the conditions we give in Eq. (10). Region II corresponds to Eq. (10) with flipped inequality signs, meaning the other superficial modality preference case where modality B is prioritized but modality A contributes more to the output. Hence, region II and III (shaded red) cover the dataset statistics where prioritization and contribution disagree and late fusion linear networks would prioritize learning the modality that contributes less to the output.

## 4  DEEP MULTIMODAL LINEAR NETWORKS

We now consider the more general case of deep multimodal linear networks and examine how the fusion layer depth $L_f$ modulates the extent of unimodal bias.

### 4.1  DEEP EARLY FUSION LINEAR NETWORK

Similar to two-layer early fusion networks, deep early fusion networks learn from both modalities simultaneously, as shown in Fig. 5a (purple curve). Saxe et al. (2014; 2019); Advani et al. (2020) have shown that for deep linear networks, depth slows down learning but does not qualitatively change the dynamics compared to two-layer linear networks. The weights associated with all input modalities escape from the initial zero fixed point $\mathcal{M}_0$ and converge to the pseudo-inverse solution fixed point in $\mathcal{M}_*$ in one transitional period.

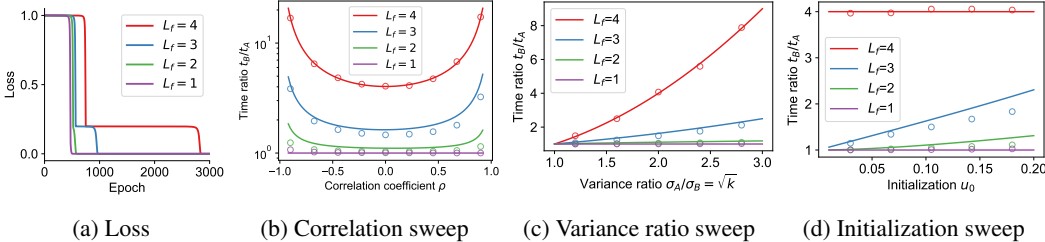

(a) Loss      (b) Correlation sweep      (c) Variance ratio sweep      (d) Initialization sweep

Figure 5: Duration of the unimodal phase in deep multimodal linear networks. Four-layer linear networks with fusion layer at $L_f = 1, 2, 3, 4$ are trained to fit $y = \boldsymbol{x}_A + \boldsymbol{x}_B$. (a) An example of loss trajectories when input covariance matrix $\boldsymbol{\Sigma} = \text{diag}(4, 1)$. (b,c,d) Lines are theory; circles are simulations of four-layer linear networks with different fusion layer depth. Variances of $\boldsymbol{x}_A, \boldsymbol{x}_B$ and their correlation coefficient are notated as $\sigma_A, \sigma_B$ and $\rho$ respectively. (b) Correlation coefficient sweep with $\sigma_A/\sigma_B = 2$ and initialization scale $u_0 = 0.1$. (c) Variance ratio sweep with $\rho = 0, u_0 = 0.1$. Note that $\sigma_A/\sigma_B = \sqrt{k}$ when $\rho = 0$. (d) Initialization scale sweep with $\sigma_A/\sigma_B = 2, \rho = 0$.

## 4.2 DEEP INTERMEDIATE AND LATE FUSION LINEAR NETWORK

Deep intermediate and late fusion linear networks learn the two modalities with two separate transitions, as shown in Fig. 5a; this is similar to what happens in two-layer late fusion networks. Due to the common terms in Eqs. (3) and (4) that govern the weight dynamics, the two manifolds of fixed points, $\mathcal{M}_0, \mathcal{M}_*$, and the two manifolds of saddles, $\mathcal{M}_A, \mathcal{M}_B$, still exist for any $2 \leq L_f \leq L, L \geq 2$, comprising intermediate and late fusion linear networks of any configuration.

In what follows, we stick to the convention that modality A is learned first. Deep intermediate and late fusion networks start from small initialization, which is close to the zero fixed point $\mathcal{M}_0$. When modality A is learned in the first transitional period at time $t_A$, the network visits the saddle $\mathcal{M}_A$. After a unimodal phase, the network goes through the second transition at time $t_B$ to reach the global pseudo-inverse solution fixed point $\mathcal{M}_*$. Because the network is in the same manifold $\mathcal{M}_A$ during the unimodal phase, our results in Section 3.2.2 on the mis-attribution in the unimodal phase and Section 3.2.3 on superficial modality preference in two-layer late fusion networks directly carry over to deep intermediate and late fusion linear networks.

As shown in Fig. 5, the loss trajectories of networks with different $L_f \geq 2$ traverse the same plateau but they stay in the plateau for different durations. We thus quantify how the total depth $L$ and fusion layer depth $L_f$ affect the duration of the unimodal phase in Section 4.2.1.

### 4.2.1 DURATION OF THE UNIMODAL PHASE

We now calculate the duration of the unimodal phase in deep intermediate and late fusion linear networks, incorporating the new parameters $L$ and $L_f$. The input-output correlation ratio is denoted as $k = \|\boldsymbol{\Sigma}_{y\boldsymbol{x}_B}\|/\|\boldsymbol{\Sigma}_{y\boldsymbol{x}_A}\| \in (0, 1)$. We derive the expression of the time ratio through leading order approximation in Appendix D. For $2 < L_f \leq L$, the time ratio is

$$\frac{t_B}{t_A} = 1 + \frac{(\|\boldsymbol{\Sigma}_{y\boldsymbol{x}_A}\| - \|\boldsymbol{\Sigma}_{y\boldsymbol{x}_B}\|)u_0^{L-L_f}}{(L_f - 2)\|\boldsymbol{\Sigma}_{y\boldsymbol{x}_A}\boldsymbol{\Sigma}_A^{-1}\|^{1-\frac{L_f}{L}} \left\|\boldsymbol{\Sigma}_{y\boldsymbol{x}_B} - \boldsymbol{\Sigma}_{y\boldsymbol{x}_A}\boldsymbol{\Sigma}_A^{-1}\boldsymbol{\Sigma}_{AB}\right\|} I(L, L_f)^{-1}, \quad (11)$$

where

$$I(L, L_f) = \int_1^\infty x^{1-L} \left[1 + \left(k + (1-k)\, x^{L_f - 2}\right)^{\frac{2}{2-L_f}}\right]^{\frac{L_f - L}{2}} dx\,. \quad (12)$$

For $L_f = 2$, the expression is slightly different; see Appendix D.3. As shown in Fig. 5, the theoretical prediction captures the trend that a deeper fusion layer $L_f$, a larger input-output correlation ratio $k$, and stronger correlations $\boldsymbol{\Sigma}_{AB}$ between input modalities all prolong the duration of the unimodal phase. The qualitative influence of dataset statistics on the time ratio in intermediate and late fusion networks is consistent with what we have seen in two-layer late fusion networks.

We now look into the influence of the fusion layer depth. By setting $L_f = L$ in Eq. (11), we find that the time ratio in deep late fusion networks reduces to the same expression as the two-layer late

fusion case in Eq. (8), which only involves dataset statistics but not the depth of the network or the initialization, since depth slows down the learning of both modalities by the same factor. In intermediate fusion linear networks, the time ratio is smaller than that in late fusion networks, with a smaller ratio with a shallower fusion layer. In intermediate fusion linear networks, learning one modality changes the weights in its associated pre-fusion layers and the shared post-fusion layers. At time $t_A$, the pre-fusion layer weights of modality A and the shared post-fusion layer weights have escaped from the zero fixed point and grown in scale while the pre-fusion layer weights of modality B have not. During the unimodal phase, the shared post-fusion layer weights and the correlation between modality B and the output together drive the pre-fusion layer weights of modality B to escape from the zero fixed point. Thus having more shared post-fusion layers makes learning one modality more helpful for learning the other, shortening the unimodal phase. In essence, an early fusion point allows the weaker modality to benefit from the stronger modality's learning in the post-fusion layers.

We also note that the initialization scale affects the time ratio in intermediate fusion networks as demonstrated in Fig. 5d. Even amongst cases that all fall into the rich feature learning regime, the initialization scale has an effect on the time ratio, with a larger time ratio for a larger initialization scale. In Fig. 5d, the simulations (circles) slightly deviate from theoretical predictions (lines) because our theoretical prediction is derived with small initialization and thus is less accurate for larger initialization. Nonetheless, the monotonic trend is well captured.

In summary, a deeper fusion layer, a larger input-output correlation ratio, stronger correlations between input modalities, and sometimes a smaller initialization scale all prolong the unimodal phase in the joint training of deep multimodal linear networks with small initialization.

## 5 DISCUSSION

We investigated the duration of the unimodal phase, mis-attribution, and superficial modality preference in deep intermediate and late fusion linear networks. We empirically find that our results, derived for linear networks, carry over to two-layer ReLU networks when the target task is linear, which aligns with the intuitions from a line of studies on two-layer ReLU networks (Sarussi et al., 2021; Phuong & Lampert, 2021; Min et al., 2023; Timor et al., 2023). We simulate and present the duration of the unimodal phase and the amount of mis-attribution in two-layer late fusion ReLU networks in Figs. 3b and 3c (crosses), which closely follow the theoretical predictions and results of their linear counterparts. The loss and weights trajectories shown in Fig. 10 are also qualitatively the same as the trajectories in two-layer linear networks in Fig. 2, except that learning is about two times slower and the converged total weights are two times larger.

In practice, multimodal data is correlated and heterogeneous. We explained the unimodal bias induced by correlation in deep linear networks. More generally, the heterogeneity in data and nonlinearity in neural networks can induce behaviors that do not arise in linear networks and linear tasks, which are out of the scope of this paper. Nonetheless, we present a simple nonlinear example in the hope of inspiring future work. Consider learning $y = x_A + \text{XOR}(x_B)$, where $x_A \in \mathbb{R}, x_B \in \{[1, 1], [1, -1], [-1, 1], [-1, -1]\}$ and $\text{XOR}(x_B)$ refers to performing XOR to the two dimensions of $x_B$. We observe that two-layer late fusion ReLU networks always learn this task successfully, forming the four perpendicular XOR features perfectly as shown in Figs. 11b, 11d and 11f. However, two-layer early fusion ReLU networks do not learn consistent XOR features and can even fail to learn this task when the variance of $x_A$ is large so that exploiting the linear modality causes fatal perturbations to extracting features from the XOR modality as shown in Figs. 11a, 11c and 11e. In this nonlinear example, late fusion networks are advantageous in terms of extracting heterogeneous features from each input modality.

We hypothesize that the practical choice of the fusion layer depth is a trade-off between alleviating unimodal bias and learning unimodal features. A shallower fusion layer helps alleviate unimodal bias because modalities can cooperate reciprocally to learn the synergistic computation. Meanwhile, a deeper fusion layer helps unimodal feature learning because the network can operate more independently to learn the unique computation of extracting heterogeneous features from each modality. We hope our work contributes to a better understanding of this tradeoff, ultimately leading to more systematic architectural choices and improved multimodal learning algorithms.

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

## APPENDIX

## A    GRADIENT DESCENT IN DEEP MULTIMODAL LINEAR NETWORKS

We derive the full-batch gradient descent dynamics in deep multimodal linear networks with learning rate $\eta$. In pre-fusion layers $1 \leq l \leq L_f$, the gradient update is

$$
\Delta \boldsymbol{W}_{\mathrm{A}}^l = -\eta \frac{\partial \mathcal{L}}{\partial \hat{y}} \frac{\partial \hat{y}}{\partial \boldsymbol{W}_{\mathrm{A}}^l}
$$

$$
= \eta \left( \prod_{j=L_f+1}^{L} \boldsymbol{W}^j \prod_{i=l+1}^{L_f} \boldsymbol{W}_{\mathrm{A}}^i \right)^{\top} (y - \hat{y}) \left( \prod_{i=1}^{l-1} \boldsymbol{W}_{\mathrm{A}}^i \boldsymbol{x}_{\mathrm{A}} \right)^{\top}
$$

$$
= \eta \left( \prod_{j=L_f+1}^{L} \boldsymbol{W}^j \prod_{i=l+1}^{L_f} \boldsymbol{W}_{\mathrm{A}}^i \right)^{\top} \left( y - \boldsymbol{W}_{\mathrm{A}}^{\mathrm{tot}} \boldsymbol{x}_{\mathrm{A}} - \boldsymbol{W}_{\mathrm{B}}^{\mathrm{tot}} \boldsymbol{x}_{\mathrm{B}} \right) \boldsymbol{x}_{\mathrm{A}}^{\top} \left( \prod_{i=1}^{l-1} \boldsymbol{W}_{\mathrm{A}}^i \right)^{\top}
$$

$$
= \eta \left( \prod_{j=L_f+1}^{L} \boldsymbol{W}^j \prod_{i=l+1}^{L_f} \boldsymbol{W}_{\mathrm{A}}^i \right)^{\top} \left( \boldsymbol{\Sigma}_{y\boldsymbol{x}_{\mathrm{A}}} - \boldsymbol{W}_{\mathrm{A}}^{\mathrm{tot}} \boldsymbol{\Sigma}_{\mathrm{A}} - \boldsymbol{W}_{\mathrm{B}}^{\mathrm{tot}} \boldsymbol{\Sigma}_{\mathrm{BA}} \right) \left( \prod_{i=1}^{l-1} \boldsymbol{W}_{\mathrm{A}}^i \right)^{\top} \quad (13\mathrm{a})
$$

$$
\Delta \boldsymbol{W}_{\mathrm{B}}^l = \left( \prod_{j=L_f+1}^{L} \boldsymbol{W}^j \prod_{i=l+1}^{L_f} \boldsymbol{W}_{\mathrm{B}}^i \right)^{\top} \left( \boldsymbol{\Sigma}_{y\boldsymbol{x}_{\mathrm{B}}} - \boldsymbol{W}_{\mathrm{A}}^{\mathrm{tot}} \boldsymbol{\Sigma}_{\mathrm{BA}} - \boldsymbol{W}_{\mathrm{B}}^{\mathrm{tot}} \boldsymbol{\Sigma}_{\mathrm{B}} \right) \left( \prod_{i=1}^{l-1} \boldsymbol{W}_{\mathrm{B}}^i \right)^{\top} \quad (13\mathrm{b})
$$

In post-fusion layers $L_f + 1 \leq l \leq L$,

$$
\Delta \boldsymbol{W}^l = -\eta \frac{\partial \mathcal{L}}{\partial \hat{y}} \frac{\partial \hat{y}}{\partial \boldsymbol{W}^l}
$$

$$
= \eta \left( \prod_{j=l+1}^{L} \boldsymbol{W}^j \right)^{\top} (y - \hat{y}) \left( \prod_{j=L_f+1}^{l-1} \boldsymbol{W}^j \prod_{i=1}^{L_f} \boldsymbol{W}_{\mathrm{A}}^i \boldsymbol{x}_{\mathrm{A}} \right)^{\top}
$$

$$
+ \eta \left( \prod_{j=l+1}^{L} \boldsymbol{W}^j \right)^{\top} (y - \hat{y}) \left( \prod_{j=L_f+1}^{l-1} \boldsymbol{W}^j \prod_{i=1}^{L_f} \boldsymbol{W}_{\mathrm{B}}^i \boldsymbol{x}_{\mathrm{B}} \right)^{\top}
$$

$$
= \eta \left( \prod_{j=l+1}^{L} \boldsymbol{W}^j \right)^{\top} \left( y - \boldsymbol{W}_{\mathrm{A}}^{\mathrm{tot}} \boldsymbol{x}_{\mathrm{A}} - \boldsymbol{W}_{\mathrm{B}}^{\mathrm{tot}} \boldsymbol{x}_{\mathrm{B}} \right) \boldsymbol{x}_{\mathrm{A}}^{\top} \left( \prod_{j=L_f+1}^{l-1} \boldsymbol{W}^j \prod_{i=1}^{L_f} \boldsymbol{W}_{\mathrm{A}}^i \right)^{\top}
$$

$$
+ \eta \left( \prod_{j=l+1}^{L} \boldsymbol{W}^j \right)^{\top} \left( y - \boldsymbol{W}_{\mathrm{A}}^{\mathrm{tot}} \boldsymbol{x}_{\mathrm{A}} - \boldsymbol{W}_{\mathrm{B}}^{\mathrm{tot}} \boldsymbol{x}_{\mathrm{B}} \right) \boldsymbol{x}_{\mathrm{B}}^{\top} \left( \prod_{j=L_f+1}^{l-1} \boldsymbol{W}^j \prod_{i=1}^{L_f} \boldsymbol{W}_{\mathrm{B}}^i \right)^{\top}
$$

$$
= \eta \left( \prod_{j=l+1}^{L} \boldsymbol{W}^j \right)^{\top} \left( \boldsymbol{\Sigma}_{y\boldsymbol{x}_{\mathrm{A}}} - \boldsymbol{W}_{\mathrm{A}}^{\mathrm{tot}} \boldsymbol{\Sigma}_{\mathrm{A}} - \boldsymbol{W}_{\mathrm{B}}^{\mathrm{tot}} \boldsymbol{\Sigma}_{\mathrm{BA}} \right) \left( \prod_{j=L_f+1}^{l-1} \boldsymbol{W}^j \prod_{i=1}^{L_f} \boldsymbol{W}_{\mathrm{A}}^i \boldsymbol{x}_{\mathrm{A}} \right)^{\top}
$$

$$
+ \eta \left( \prod_{j=l+1}^{L} \boldsymbol{W}^j \right)^{\top} \left( \boldsymbol{\Sigma}_{y\boldsymbol{x}_{\mathrm{B}}} - \boldsymbol{W}_{\mathrm{A}}^{\mathrm{tot}} \boldsymbol{\Sigma}_{\mathrm{AB}} - \boldsymbol{W}_{\mathrm{B}}^{\mathrm{tot}} \boldsymbol{\Sigma}_{\mathrm{B}} \right) \left( \prod_{j=L_f+1}^{l-1} \boldsymbol{W}^j \prod_{i=1}^{L_f} \boldsymbol{W}_{\mathrm{B}}^i \boldsymbol{x}_{\mathrm{B}} \right)^{\top} \quad (14)
$$

In the limit of small learning rate, the difference equations Eqs. (13) and (14) are well approximated by the differential equations Eqs. (3) and (4) in the main text.

## B  Two-Layer Early Fusion Linear Network Fixed points

A two-layer early fusion linear network is described as $\hat{y} = \boldsymbol{W}^2\boldsymbol{W}^1\boldsymbol{x}$. The gradient descent dynamics are

$$\tau\dot{\boldsymbol{W}}^1 = \boldsymbol{W}^{2\top}(\boldsymbol{\Sigma}_{y\boldsymbol{x}} - \boldsymbol{W}^2\boldsymbol{W}^1\boldsymbol{\Sigma}), \quad \tau\dot{\boldsymbol{W}}^2 = (\boldsymbol{\Sigma}_{y\boldsymbol{x}} - \boldsymbol{W}^2\boldsymbol{W}^1\boldsymbol{\Sigma})\boldsymbol{W}^{1\top}. \tag{15}$$

By setting to the gradients to zero, we find that there are two manifolds of fixed points:

$$\boldsymbol{W}^2 = \boldsymbol{0}, \boldsymbol{W}^1 = \boldsymbol{0} \quad \Rightarrow \quad \mathcal{M}_0 = \{\boldsymbol{W}|\boldsymbol{W} = \boldsymbol{0}\}, \tag{16a}$$

$$\boldsymbol{\Sigma}_{y\boldsymbol{x}} - \boldsymbol{W}^2\boldsymbol{W}^1\boldsymbol{\Sigma} = \boldsymbol{0} \quad \Rightarrow \quad \mathcal{M}_* = \{\boldsymbol{W}|\boldsymbol{W}^{\text{tot}} = \boldsymbol{\Sigma}_{y\boldsymbol{x}}\boldsymbol{\Sigma}^{-1}\}. \tag{16b}$$

## C  Two-Layer Late Fusion Linear Network Dynamics

A two-layer late fusion linear network is described as $\hat{y} = \boldsymbol{W}_{\text{A}}^2\boldsymbol{W}_{\text{A}}^1\boldsymbol{x}_{\text{A}} + \boldsymbol{W}_{\text{B}}^2\boldsymbol{W}_{\text{B}}^1\boldsymbol{x}_{\text{B}}$. The gradient descent dynamics are

$$\tau\dot{\boldsymbol{W}}_{\text{A}}^1 = \boldsymbol{W}_{\text{A}}^{2\top}(\boldsymbol{\Sigma}_{y\boldsymbol{x}_{\text{A}}} - \boldsymbol{W}_{\text{A}}^2\boldsymbol{W}_{\text{A}}^1\boldsymbol{\Sigma}_{\text{A}} - \boldsymbol{W}_{\text{B}}^2\boldsymbol{W}_{\text{B}}^1\boldsymbol{\Sigma}_{\text{BA}}), \tag{17a}$$

$$\tau\dot{\boldsymbol{W}}_{\text{A}}^2 = (\boldsymbol{\Sigma}_{y\boldsymbol{x}_{\text{A}}} - \boldsymbol{W}_{\text{A}}^2\boldsymbol{W}_{\text{A}}^1\boldsymbol{\Sigma}_{\text{A}} - \boldsymbol{W}_{\text{B}}^2\boldsymbol{W}_{\text{B}}^1\boldsymbol{\Sigma}_{\text{BA}})\boldsymbol{W}_{\text{A}}^{1\top}, \tag{17b}$$

$$\tau\dot{\boldsymbol{W}}_{\text{B}}^1 = \boldsymbol{W}_{\text{B}}^{2\top}(\boldsymbol{\Sigma}_{y\boldsymbol{x}_{\text{B}}} - \boldsymbol{W}_{\text{A}}^2\boldsymbol{W}_{\text{A}}^1\boldsymbol{\Sigma}_{\text{AB}} - \boldsymbol{W}_{\text{B}}^2\boldsymbol{W}_{\text{B}}^1\boldsymbol{\Sigma}_{\text{B}}), \tag{17c}$$

$$\tau\dot{\boldsymbol{W}}_{\text{B}}^2 = (\boldsymbol{\Sigma}_{y\boldsymbol{x}_{\text{B}}} - \boldsymbol{W}_{\text{A}}^2\boldsymbol{W}_{\text{A}}^1\boldsymbol{\Sigma}_{\text{AB}} - \boldsymbol{W}_{\text{B}}^2\boldsymbol{W}_{\text{B}}^1\boldsymbol{\Sigma}_{\text{B}})\boldsymbol{W}_{\text{B}}^{1\top}. \tag{17d}$$

### C.1  Fixed Points and Saddles

By setting the gradients in Eq. (17) to zero, we find that the two manifolds of fixed points in Eq. (5) exist in two-layer late fusion linear networks as well.

$$\begin{cases} \boldsymbol{W}_{\text{A}}^2 = \boldsymbol{0}, \boldsymbol{W}_{\text{A}}^1 = \boldsymbol{0} \\ \boldsymbol{W}_{\text{B}}^2 = \boldsymbol{0}, \boldsymbol{W}_{\text{B}}^1 = \boldsymbol{0} \end{cases} \Rightarrow \quad \mathcal{M}_0 = \{\boldsymbol{W}|\boldsymbol{W} = \boldsymbol{0}\}, \tag{18a}$$

$$\begin{cases} \boldsymbol{\Sigma}_{y\boldsymbol{x}_{\text{A}}} - \boldsymbol{W}_{\text{A}}^2\boldsymbol{W}_{\text{A}}^1\boldsymbol{\Sigma}_{\text{A}} - \boldsymbol{W}_{\text{B}}^2\boldsymbol{W}_{\text{B}}^1\boldsymbol{\Sigma}_{\text{BA}} = \boldsymbol{0} \\ \boldsymbol{\Sigma}_{y\boldsymbol{x}_{\text{B}}} - \boldsymbol{W}_{\text{A}}^2\boldsymbol{W}_{\text{A}}^1\boldsymbol{\Sigma}_{\text{AB}} - \boldsymbol{W}_{\text{B}}^2\boldsymbol{W}_{\text{B}}^1\boldsymbol{\Sigma}_{\text{B}} = \boldsymbol{0} \end{cases} \Rightarrow \quad \mathcal{M}_* = \{\boldsymbol{W}|\boldsymbol{W}^{\text{tot}} = \boldsymbol{\Sigma}_{y\boldsymbol{x}}\boldsymbol{\Sigma}^{-1}\}. \tag{18b}$$

In addition, there are two manifolds of saddles:

$$\begin{cases} \boldsymbol{\Sigma}_{y\boldsymbol{x}_{\text{A}}} - \boldsymbol{W}_{\text{A}}^2\boldsymbol{W}_{\text{A}}^1\boldsymbol{\Sigma}_{\text{A}} = \boldsymbol{0} \\ \boldsymbol{W}_{\text{B}}^2 = \boldsymbol{0}, \boldsymbol{W}_{\text{B}}^1 = \boldsymbol{0} \end{cases} \Rightarrow \quad \mathcal{M}_{\text{A}} = \{\boldsymbol{W}|\boldsymbol{W}_{\text{A}}^{\text{tot}} = \boldsymbol{\Sigma}_{y\boldsymbol{x}_{\text{A}}}\boldsymbol{\Sigma}_{\text{A}}^{-1}, \boldsymbol{W}_{\text{B}}^{\text{tot}} = \boldsymbol{0}\}, \tag{19a}$$

$$\begin{cases} \boldsymbol{W}_{\text{A}}^2 = \boldsymbol{0}, \boldsymbol{W}_{\text{A}}^1 = \boldsymbol{0} \\ \boldsymbol{\Sigma}_{y\boldsymbol{x}_{\text{B}}} - \boldsymbol{W}_{\text{B}}^2\boldsymbol{W}_{\text{B}}^1\boldsymbol{\Sigma}_{\text{B}} = \boldsymbol{0} \end{cases} \Rightarrow \quad \mathcal{M}_{\text{B}} = \{\boldsymbol{W}|\boldsymbol{W}_{\text{A}}^{\text{tot}} = \boldsymbol{0}, \boldsymbol{W}_{\text{B}}^{\text{tot}} = \boldsymbol{\Sigma}_{y\boldsymbol{x}_{\text{B}}}\boldsymbol{\Sigma}_{\text{B}}^{-1}\}. \tag{19b}$$

We plot the four manifolds $\mathcal{M}_0, \mathcal{M}_*, \mathcal{M}_{\text{A}}, \mathcal{M}_{\text{B}}$ in Fig. 6 in a scalar case introduced in Fig. 2d.

### C.2  A Solvable Simple Case

If the two modalities are not correlated with each other ($\boldsymbol{\Sigma}_{\text{AB}} = \boldsymbol{0}$) and have white correlations ($\boldsymbol{\Sigma}_{\text{A}} = \sigma_{\text{A}}^2\boldsymbol{I}, \boldsymbol{\Sigma}_{\text{B}} = \sigma_{\text{B}}^2\boldsymbol{I}$) the dynamics is equivalent to two separately trained unimodal two-layer linear networks with whitened input, whose solution has been derived in Saxe et al. (2014). The time course solution of total weights is

$$\boldsymbol{W}_{\text{A}}^{\text{tot}}(t) = u_{\text{A}}(t)\sigma_{\text{A}}^{-2}\boldsymbol{\Sigma}_{y\boldsymbol{x}_{\text{A}}}, \, u_{\text{A}}(t) = \left[\left(\frac{\|\boldsymbol{\Sigma}_{y\boldsymbol{x}_{\text{A}}}\|}{\sigma_{\text{A}}^2 u_{\text{A}}(0)} - 1\right)e^{-2\|\boldsymbol{\Sigma}_{y\boldsymbol{x}_{\text{A}}}\|\frac{t}{\tau}} + 1\right]^{-1}, \tag{20a}$$

$$\boldsymbol{W}_{\text{B}}^{\text{tot}}(t) = u_{\text{B}}(t)\sigma_{\text{B}}^{-2}\boldsymbol{\Sigma}_{y\boldsymbol{x}_{\text{B}}}, \, u_{\text{B}}(t) = \left[\left(\frac{\|\boldsymbol{\Sigma}_{y\boldsymbol{x}_{\text{B}}}\|}{\sigma_{\text{B}}^2 u_{\text{B}}(0)} - 1\right)e^{-2\|\boldsymbol{\Sigma}_{y\boldsymbol{x}_{\text{B}}}\|\frac{t}{\tau}} + 1\right]^{-1}. \tag{20b}$$

The total weights for both modalities only evolve in scale along the pseudo-inverse solution direction. The scale variables $u_{\text{A}}(t), u_{\text{B}}(t)$ go through the same sigmoidal growth while modality A grows approximately $\|\boldsymbol{\Sigma}_{y\boldsymbol{x}_{\text{A}}}\|/\|\boldsymbol{\Sigma}_{y\boldsymbol{x}_{\text{B}}}\|$ times faster.

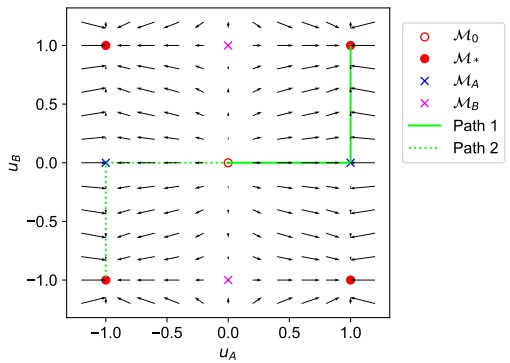

Figure 6: Phase portrait corresponding to learning dynamics in Fig. 2d. The horizontal axis $u_A$ is the Frobenius norm of a layer in branch A as defined in Eq. (22); and similarly for the vertical axis $u_B$. Two-layer late fusion linear networks initialized with small weights either go along the solid or the dotted green path to reach the solid red dot stable fixed point in $\mathcal{M}_*$ at convergence. Via either path, the network visits and lingers in a saddle in manifold $\mathcal{M}_A$, giving rise to the unimodal phase.

### C.3 GENERAL CASES

In the general case of having arbitrary correlation matrices, the analytical solution in Eq. (20) does not hold since the total weights not only grow in one fixed direction but also rotate. We then focus on the early phase dynamics to compute the duration of the unimodal phase.

### C.3.1 WHICH MODALITY IS LEARNED FIRST AND WHEN?

We assume that the modality learned first is modality A and specify the time $t_A$ and the condition for prioritizing modality A in the following. During time $0$ to $t_A$, the total weights of both modalities have not started to grow appreciably. Hence, the dynamics during time $0 \sim t_A$ in leading order approximation are

$$\tau \dot{W}_A^1 = W_A^{2\top} \Sigma_{y x_A}, \ \tau \dot{W}_A^2 = \Sigma_{y x_A} W_A^{1\top}; \tag{21a}$$

$$\tau \dot{W}_B^1 = W_B^{2\top} \Sigma_{y x_B}, \ \tau \dot{W}_B^2 = \Sigma_{y x_B} W_B^{1\top}. \tag{21b}$$

The first-layer weights $W_A^1, W_B^1$ align with $\Sigma_{y x_A}, \Sigma_{y x_B}$ respectively in the early phase when their scale has not grown appreciably (Atanasov et al., 2022). We further have the balancing property (Du et al., 2018; Ji & Telgarsky, 2019) between the two layers of modality A

$$W_A^1 W_A^{1\top} = W_A^{2\top} W_A^2 \quad \Rightarrow \quad \|W_A^1\|_F = \|W_A^2\| \stackrel{\text{def}}{=} u_A. \tag{22}$$

We conduct the following change of variable

$$W_A^1 = u_A(t) r_A^1 \frac{\Sigma_{y x_A}}{\|\Sigma_{y x_A}\|}, \ W_A^2 = u_A(t) r_A^{1\top}, \tag{23}$$

where $r_A^1$ is a fixed unit norm column vector representing the freedom in the hidden layer and $u_A$ is a scalar representing the norm of the two balancing layers (Advani et al., 2020). Substituting the variables, we can re-write the dynamics of $W_A^2$ and obtain an ordinary differential equation about $u_A$:

$$\tau \dot{u}_A r_A^{1\top} = \Sigma_{y x_A} \frac{\Sigma_{y x_A}^\top}{\|\Sigma_{y x_A}\|} r_A^{1\top} u_A \quad \Rightarrow \quad \tau \dot{u}_A = \|\Sigma_{y x_A}\| u_A. \tag{24}$$

Separating variables and integrating both sides, we get

$$t = \tau \|\Sigma_{y x_A}\|^{-1} (\ln u_A - \ln u_0), \tag{25}$$

where $u_0$ denotes $u_0 = u_A(0)$. Because the initialization $u_0$ is very small compared to $\|\Sigma_{y x_A}\|$, the time that $u_A$ grows to be comparable with $\|\Sigma_{y x_A}\|$ is

$$t_A \approx \tau \|\Sigma_{y x_A}\|^{-1} \ln \frac{1}{u_0}, \tag{26}$$

From Eq. (26), we can see that the condition for modality A to be learned first is that $\|\mathbf{\Sigma}_{y\boldsymbol{x}_A}\| > \|\mathbf{\Sigma}_{y\boldsymbol{x}_B}\|$.

$$\tau\|\mathbf{\Sigma}_{y\boldsymbol{x}_A}\|^{-1}\ln\frac{1}{u_0} < \tau\|\mathbf{\Sigma}_{y\boldsymbol{x}_B}\|^{-1}\ln\frac{1}{u_0} \quad \Leftrightarrow \quad \|\mathbf{\Sigma}_{y\boldsymbol{x}_A}\| > \|\mathbf{\Sigma}_{y\boldsymbol{x}_B}\|. \tag{27}$$

### C.3.2 When is the Second Modality Learned?

We next calculate the time $t_B$ when modality B is learned. During time $0$ to $t_A$, the weight dynamics of modality B takes the same form as modality A as described in Eq. (21b). Because the layers in the modality B branch are balanced and $\boldsymbol{W}_B^1$ aligns with the rank-one direction $\mathbf{\Sigma}_{y\boldsymbol{x}_B}$, we again change variables and re-write the dynamics of $\boldsymbol{W}_B^2$, obtaining an ordinary differential equation about the norm of the two balanced layers $u_B = \|\boldsymbol{W}_B^1\|_F = \|\boldsymbol{W}_B^2\|$:

$$\tau\dot{u}_B = \|\mathbf{\Sigma}_{y\boldsymbol{x}_B}\|u_B \quad \Rightarrow \quad t = \tau\|\mathbf{\Sigma}_{y\boldsymbol{x}_B}\|^{-1}(\ln u_B - \ln u_0), \ t\in[0, t_A). \tag{28}$$

We assume $u_B(0) = u_0$, since the initialization is small and the number of hidden neurons in the modality B branch is of the same order as modality A. We plug $t_A$ from Eq. (26) into Eq. (28) and get

$$\ln u_B(t_A) = \left(1 - \|\mathbf{\Sigma}_{y\boldsymbol{x}_A}\|^{-1}\|\mathbf{\Sigma}_{y\boldsymbol{x}_B}\|\right)\ln u_0. \tag{29}$$

We then look into the dynamics during the time lag from $t_A$ to $t_B$. During time $t_A$ to $t_B$, the weights of modality B are still small and negligible to leading ordering approximation. Meanwhile, the weights of modality A have grown to be $\boldsymbol{W}_A^{tot} = \mathbf{\Sigma}_{y\boldsymbol{x}_A}\mathbf{\Sigma}_A^{-1}$, which changes the dynamics of modality B when the cross-covariance $\mathbf{\Sigma}_{AB} \neq \mathbf{0}$. Taking this into consideration, the dynamics of modality B during $t_A$ to $t_B$ is

$$\tau\dot{\boldsymbol{W}}_B^1 = \boldsymbol{W}_B^{2\top}\widetilde{\mathbf{\Sigma}}_{y\boldsymbol{x}_B}, \ \tau\dot{\boldsymbol{W}}_B^2 = \widetilde{\mathbf{\Sigma}}_{y\boldsymbol{x}_B}\boldsymbol{W}_B^{1\top}, \ t\in(t_A, t_B), \tag{30}$$

where $\widetilde{\mathbf{\Sigma}}_{y\boldsymbol{x}_B} = \mathbf{\Sigma}_{y\boldsymbol{x}_B} - \mathbf{\Sigma}_{y\boldsymbol{x}_A}\mathbf{\Sigma}_A^{-1}\mathbf{\Sigma}_{AB}$. The first-layer weights $\boldsymbol{W}_B^1$ rapidly rotate from $\mathbf{\Sigma}_{y\boldsymbol{x}_B}$ to $\widetilde{\mathbf{\Sigma}}_{y\boldsymbol{x}_B}$ at time $t_A$ and continue to evolve along $\widetilde{\mathbf{\Sigma}}_{y\boldsymbol{x}_B}$ during $t_A$ to $t_B$. Through the same manner of changing variables, we obtain the ordinary differential equation about $u_B$ during $t_A$ to $t_B$:

$$\tau\dot{u}_B = \|\widetilde{\mathbf{\Sigma}}_{y\boldsymbol{x}_B}\|u_B \quad \Rightarrow \quad t - t_A = \tau\|\widetilde{\mathbf{\Sigma}}_{y\boldsymbol{x}_B}\|^{-1}(\ln u_B - \ln u_B(t_A)), \ t\in(t_A, t_B). \tag{31}$$

Plugging in $u_B(t_A)$ obtained in Eq. (29), we get the time when $u_B$ grows to be comparable with $\|\mathbf{\Sigma}_{y\boldsymbol{x}_B}\|$:

$$t_B \approx t_A - \tau\|\widetilde{\mathbf{\Sigma}}_{y\boldsymbol{x}_B}\|^{-1}\ln u_B(t_A) \approx t_A + \tau\frac{1 - \|\mathbf{\Sigma}_{y\boldsymbol{x}_A}\|^{-1}\|\mathbf{\Sigma}_{y\boldsymbol{x}_B}\|}{\|\widetilde{\mathbf{\Sigma}}_{y\boldsymbol{x}_B}\|}\ln\frac{1}{u_0}. \tag{32}$$

Dividing Eq. (32) by Eq. (26), we obtain the time ratio Eq. (8) in the main text:

$$\frac{t_B}{t_A} = 1 + \frac{\|\mathbf{\Sigma}_{y\boldsymbol{x}_A}\| - \|\mathbf{\Sigma}_{y\boldsymbol{x}_B}\|}{\left\|\mathbf{\Sigma}_{y\boldsymbol{x}_B} - \mathbf{\Sigma}_{y\boldsymbol{x}_A}\mathbf{\Sigma}_A^{-1}\mathbf{\Sigma}_{AB}\right\|}. \tag{33}$$

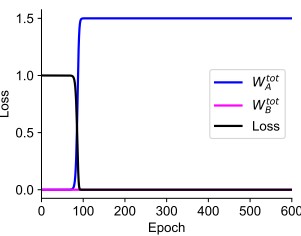

Figure 7: Loss and weight trajectories in a two-layer late fusion network when modalities have collinearity. Inputs $\boldsymbol{x}_A$ and $\boldsymbol{x}_B$ are scalars with covariance matrix $\mathbf{\Sigma} = [4, 2; 2, 1]$. The target output is generated as $y = \boldsymbol{x}_A + \boldsymbol{x}_B$.

Note that if $\boldsymbol{x}_A$ and $\boldsymbol{x}_B$ have collinearity, the denominator in Eq. (33) is zero and the time ratio is $\infty$. As shown in Fig. 7, the late fusion network learns to fit the output only with modality A and modality B will never be learned.

# D  DEEP INTERMEDIATE & LATE FUSION LINEAR NETWORKS

## D.1  BALANCING PROPERTIES

The full dynamics of deep intermediate and late fusion linear networks are given in Eqs. (3) and (4). Between pre-fusion layers and between post-fusion layers, the standard balancing property (Du et al., 2018; Ji & Telgarsky, 2019) holds true:

$$\boldsymbol{W}_{\mathrm{A}}^1 \boldsymbol{W}_{\mathrm{A}}^{1^\top} = \boldsymbol{W}_{\mathrm{A}}^2 \boldsymbol{W}_{\mathrm{A}}^{2^\top} = \cdots = \boldsymbol{W}_{\mathrm{A}}^{L_f} \boldsymbol{W}_{\mathrm{A}}^{L_f^\top}, \tag{34a}$$

$$\boldsymbol{W}_{\mathrm{B}}^1 \boldsymbol{W}_{\mathrm{B}}^{1^\top} = \boldsymbol{W}_{\mathrm{B}}^2 \boldsymbol{W}_{\mathrm{B}}^{2^\top} = \cdots = \boldsymbol{W}_{\mathrm{B}}^{L_f} \boldsymbol{W}_{\mathrm{B}}^{L_f^\top}, \tag{34b}$$

$$\boldsymbol{W}^{L_f+1} \boldsymbol{W}^{L_f+1^\top} = \boldsymbol{W}^{L_f+2} \boldsymbol{W}^{L_f+2^\top} = \cdots = \boldsymbol{W}^{L^\top} \boldsymbol{W}^L. \tag{34c}$$

Between a pre-fusion layer and a post-fusion layer, the balancing condition takes a slightly different form:

$$\boldsymbol{W}_{\mathrm{A}}^{L_f} \boldsymbol{W}_{\mathrm{A}}^{L_f^\top} + \boldsymbol{W}_{\mathrm{B}}^{L_f} \boldsymbol{W}_{\mathrm{B}}^{L_f^\top} = \boldsymbol{W}^{L_f+1} \boldsymbol{W}^{L_f+1^\top}. \tag{35}$$

Based on the standard balancing properties in Eq. (34), we have equal norm between pre-fusion layers and between post-fusion layers.

$$\|\boldsymbol{W}_{\mathrm{A}}^1\|_{\mathrm{F}} = \|\boldsymbol{W}_{\mathrm{A}}^2\|_{\mathrm{F}} = \cdots = \|\boldsymbol{W}_{\mathrm{A}}^{L_f}\|_{\mathrm{F}} \overset{\text{def}}{=} u_{\mathrm{A}}, \tag{36a}$$

$$\|\boldsymbol{W}_{\mathrm{B}}^1\|_{\mathrm{F}} = \|\boldsymbol{W}_{\mathrm{B}}^2\|_{\mathrm{F}} = \cdots = \|\boldsymbol{W}_{\mathrm{B}}^{L_f}\|_{\mathrm{F}} \overset{\text{def}}{=} u_{\mathrm{B}}, \tag{36b}$$

$$\|\boldsymbol{W}^{L_f+1}\|_{\mathrm{F}} = \|\boldsymbol{W}^{L_f+2}\|_{\mathrm{F}} = \cdots = \|\boldsymbol{W}^L\| \overset{\text{def}}{=} u. \tag{36c}$$

Based on Eq. (35), we have the balancing property between the norm of a pre-fusion layer and a post-fusion layer.

$$u_{\mathrm{A}}^2 + u_{\mathrm{B}}^2 = u^2. \tag{37}$$

In addition to the balancing norm, we infer from the balancing properties that post-fusion layers have rank-one structures as in standard linear networks (Ji & Telgarsky, 2019; Atanasov et al., 2022). The pre-fusion layers are not necessarily rank-one due to the different balancing property in Eq. (35). However, guided by empirical observations, we make the ansatz that the weights in pre-fusion layers are also rank-one, which will enable us to conduct the change of variables similar to what we have done in Appendix C.3.

## D.2  TIME RATIO FOR $L_f \neq 2$

### D.2.1  WHICH MODALITY IS LEARNED FIRST AND WHEN?

We adopt the convention that the modality learned first is modality A. During time 0 to $t_{\mathrm{A}}$, the dynamics in leading order approximation are

$$\tau \dot{\boldsymbol{W}}_{\mathrm{A}}^l = \left(\prod_{j=L_f+1}^{L} \boldsymbol{W}^j \prod_{i=l+1}^{L_f} \boldsymbol{W}_{\mathrm{A}}^i\right)^\top \boldsymbol{\Sigma}_{y\boldsymbol{x}_{\mathrm{A}}} \left(\prod_{i=1}^{l-1} \boldsymbol{W}_{\mathrm{A}}^i\right)^\top, \ 1 \leq l \leq L_f, \tag{38a}$$

$$\tau \dot{\boldsymbol{W}}_{\mathrm{B}}^l = \left(\prod_{j=L_f+1}^{L} \boldsymbol{W}^j \prod_{i=l+1}^{L_f} \boldsymbol{W}_{\mathrm{B}}^i\right)^\top \boldsymbol{\Sigma}_{y\boldsymbol{x}_{\mathrm{B}}} \left(\prod_{i=1}^{l-1} \boldsymbol{W}_{\mathrm{B}}^i\right)^\top, \ 1 \leq l \leq L_f, \tag{38b}$$

$$\tau \dot{\boldsymbol{W}}^l = \left(\prod_{j=l+1}^{L} \boldsymbol{W}^j\right)^\top \boldsymbol{\Sigma}_{y\boldsymbol{x}_{\mathrm{A}}} \left(\prod_{j=L_f+1}^{l-1} \boldsymbol{W}^j \prod_{i=1}^{L_f} \boldsymbol{W}_{\mathrm{A}}^i\right)^\top$$
$$+ \left(\prod_{j=l+1}^{L} \boldsymbol{W}^j\right)^\top \boldsymbol{\Sigma}_{y\boldsymbol{x}_{\mathrm{B}}} \left(\prod_{j=L_f+1}^{l-1} \boldsymbol{W}^j \prod_{i=1}^{L_f} \boldsymbol{W}_{\mathrm{B}}^i\right)^\top, \ L_f+1 \leq l \leq L. \tag{38c}$$

We make the following change of variables

$$\boldsymbol{W}_{\mathrm{A}}^1 = u_{\mathrm{A}}(t)\boldsymbol{r}_{\mathrm{A}}^1 \frac{\boldsymbol{\Sigma}_{y\boldsymbol{x}_{\mathrm{A}}}}{\|\boldsymbol{\Sigma}_{y\boldsymbol{x}_{\mathrm{A}}}\|}, \; \boldsymbol{W}_{\mathrm{B}}^1 = u_{\mathrm{B}}(t)\boldsymbol{r}_{\mathrm{B}}^1 \frac{\boldsymbol{\Sigma}_{y\boldsymbol{x}_{\mathrm{B}}}}{\|\boldsymbol{\Sigma}_{y\boldsymbol{x}_{\mathrm{B}}}\|}; \tag{39a}$$

$$\boldsymbol{W}_{\mathrm{A}}^i = u_{\mathrm{A}}(t)\boldsymbol{r}_{\mathrm{A}}^i \boldsymbol{r}_{\mathrm{A}}^{i-1\top}, \; \boldsymbol{W}_{\mathrm{B}}^i = u_{\mathrm{B}}(t)\boldsymbol{r}_{\mathrm{B}}^i \boldsymbol{r}_{\mathrm{B}}^{i-1\top}, \; 2 \le i \le L_f - 1; \tag{39b}$$

$$\boldsymbol{W}_{\mathrm{A}}^{L_f} = u_{\mathrm{A}}(t)\boldsymbol{r}^{L_f}\boldsymbol{r}_{\mathrm{A}}^{L_f-1\top}, \; \boldsymbol{W}_{\mathrm{B}}^{L_f} = u_{\mathrm{B}}(t)\boldsymbol{r}^{L_f}\boldsymbol{r}_{\mathrm{B}}^{L_f-1\top}; \tag{39c}$$

$$\boldsymbol{W}^j = u(t)\boldsymbol{r}^j \boldsymbol{r}^{j-1\top}, \; L_f + 1 \le j \le L - 1; \tag{39d}$$

$$\boldsymbol{W}^L = u(t)\boldsymbol{r}^{L\top}. \tag{39e}$$

where all $\boldsymbol{r}$ are fixed unit norm column vectors representing the freedom in hidden layers. By substituing Eq. (39) into Eq. (38), we reduce the dynamics during time 0 to $t_{\mathrm{A}}$ to a three-dimensional dynamical system about $u_{\mathrm{A}}, u_{\mathrm{B}}$ and $u$:

$$\tau \dot{u}_{\mathrm{A}} = \|\boldsymbol{\Sigma}_{y\boldsymbol{x}_{\mathrm{A}}}\| u_{\mathrm{A}}^{L_f-1} u^{L-L_f}, \tag{40a}$$

$$\tau \dot{u}_{\mathrm{B}} = \|\boldsymbol{\Sigma}_{y\boldsymbol{x}_{\mathrm{B}}}\| u_{\mathrm{B}}^{L_f-1} u^{L-L_f}, \tag{40b}$$

$$\tau \dot{u} = \|\boldsymbol{\Sigma}_{y\boldsymbol{x}_{\mathrm{A}}}\| u_{\mathrm{A}}^{L_f} u^{L-L_f-1} + \|\boldsymbol{\Sigma}_{y\boldsymbol{x}_{\mathrm{B}}}\| u_{\mathrm{B}}^{L_f} u^{L-L_f-1}. \tag{40c}$$

We divide Eq. (40b) by Eq. (40a) and reveal an equality between the two branches of a pre-fusion layer:

$$\frac{du_{\mathrm{B}}}{du_{\mathrm{A}}} = \frac{\|\boldsymbol{\Sigma}_{y\boldsymbol{x}_{\mathrm{B}}}\| u_{\mathrm{B}}^{L_f-1}}{\|\boldsymbol{\Sigma}_{y\boldsymbol{x}_{\mathrm{A}}}\| u_{\mathrm{A}}^{L_f-1}} \;\; \Rightarrow \;\; \frac{u_{\mathrm{B}}^{2-L_f} - u_0^{2-L_f}}{u_{\mathrm{A}}^{2-L_f} - u_0^{2-L_f}} = \frac{\|\boldsymbol{\Sigma}_{y\boldsymbol{x}_{\mathrm{B}}}\|}{\|\boldsymbol{\Sigma}_{y\boldsymbol{x}_{\mathrm{A}}}\|} \stackrel{\text{def}}{=} k, \; L_f \neq 2, \tag{41}$$

We assume $L_f \neq 2$ in this section and handle the case of $L_f = 2$ in Appendix D.3. By utilizing two equality properties in Eq. (41) and Eq. (37), we reduce Eq. (40a) to an ordinary differential equation:

$$\tau \dot{u}_{\mathrm{A}} = \|\boldsymbol{\Sigma}_{y\boldsymbol{x}_{\mathrm{A}}}\| u_{\mathrm{A}}^{L_f-1} \left[ u_{\mathrm{A}}^2 + \left( k u_{\mathrm{A}}^{2-L_f} + (1-k) u_0^{2-L_f} \right)^{\frac{2}{2-L_f}} \right]^{\frac{L-L_f}{2}}. \tag{42}$$

The ordinary differential equation Eq. (42) is separable despite being cumbersome. By separating variables and integrating both sides, we can write time $t$ as a function of $u_{\mathrm{A}}$:

$$t = \int_{u_0}^{u_{\mathrm{A}}} x^{1-L_f} \left[ x^2 + \left( k x^{2-L_f} + (1-k) u_0^{2-L_f} \right)^{\frac{2}{2-L_f}} \right]^{\frac{L_f-L}{2}} dx. \tag{43}$$

The time when $u_{\mathrm{A}}$ grows to be comparable with $\|\boldsymbol{\Sigma}_{y\boldsymbol{x}_{\mathrm{A}}}\|$, for instance $\|\boldsymbol{\Sigma}_{y\boldsymbol{x}_{\mathrm{A}}}\|/2$, is

$$t_{\mathrm{A}} \approx \tau \|\boldsymbol{\Sigma}_{y\boldsymbol{x}_{\mathrm{A}}}\|^{-1} \int_{u_0}^{\|\boldsymbol{\Sigma}_{y\boldsymbol{x}_{\mathrm{A}}}\|/2} u_{\mathrm{A}}^{1-L_f} \left[ u_{\mathrm{A}}^2 + \left( k u_{\mathrm{A}}^{2-L_f} + (1-k) u_0^{2-L_f} \right)^{\frac{2}{2-L_f}} \right]^{\frac{L_f-L}{2}} du_{\mathrm{A}}$$

$$\approx \tau \|\boldsymbol{\Sigma}_{y\boldsymbol{x}_{\mathrm{A}}}\|^{-1} u_0^{2-L} \int_1^\infty x^{1-L} \left[ 1 + \left( k + (1-k) x^{L_f-2} \right)^{\frac{2}{2-L_f}} \right]^{\frac{L_f-L}{2}} dx. \tag{44}$$

From Eq. (44), we find that the condition for modality A to be learned first in deep intermediate and late fusion linear networks is the same as that for two-layer late fuison linear networks:

$$\tau \|\boldsymbol{\Sigma}_{y\boldsymbol{x}_{\mathrm{A}}}\|^{-1} u_0^{2-L_f} I(L, L_f) < \tau \|\boldsymbol{\Sigma}_{y\boldsymbol{x}_{\mathrm{B}}}\|^{-1} u_0^{2-L_f} I(L, L_f) \;\; \Leftrightarrow \;\; \frac{\|\boldsymbol{\Sigma}_{y\boldsymbol{x}_{\mathrm{B}}}\|}{\|\boldsymbol{\Sigma}_{y\boldsymbol{x}_{\mathrm{A}}}\|} \equiv k \in (0, 1), \tag{45}$$

where we use $I(L, L_f)$ to denote the integral

$$I(L, L_f) = \int_1^\infty x^{1-L} \left[ 1 + \left( k + (1-k) x^{L_f-2} \right)^{\frac{2}{2-L_f}} \right]^{\frac{L_f-L}{2}} dx. \tag{46}$$

### D.2.2 WHEN IS THE SECOND MODALITY LEARNED?

We next compute the time $t_{\text{B}}$ when modality B is learned. During time $t_{\text{A}}$ to $t_{\text{B}}$, the network is in manifold $\mathcal{M}_{\text{A}}$ where

$$\boldsymbol{W}_{\text{A}}^{\text{tot}} = \boldsymbol{\Sigma}_{y\boldsymbol{x}_{\text{A}}}\boldsymbol{\Sigma}_{\text{A}}^{-1} \quad \Rightarrow \quad u_{\text{A}} = u = \left\|\boldsymbol{\Sigma}_{y\boldsymbol{x}_{\text{A}}}\boldsymbol{\Sigma}_{\text{A}}^{-1}\right\|^{\frac{1}{L}}, \ t \in (t_{\text{A}}, t_{\text{B}}). \tag{47}$$

We plug $u_{\text{A}}(t_{\text{A}})$, which is very large compared to $u_0$, into Eq. (41) and obtain $u_{\text{B}}(t_{\text{A}})$

$$u_{\text{B}}(t_{\text{A}}) = \left[u_{\text{A}}(t_{\text{A}})^{2-L_f} + (1-k)u_0^{2-L_f}\right]^{\frac{1}{2-L_f}} \approx (1-k)^{\frac{1}{2-L_f}} u_0. \tag{48}$$

We then look into the dynamics during the time lag $t_{\text{A}}$ to $t_{\text{B}}$. Substituting $\boldsymbol{W}_{\text{A}}^{\text{tot}} = \boldsymbol{\Sigma}_{y\boldsymbol{x}_{\text{A}}}\boldsymbol{\Sigma}_{\text{A}}^{-1}$ into Eq. (3b), we get

$$\tau\dot{\boldsymbol{W}}_{\text{B}}^l = \left(\prod_{j=L_f+1}^{L} \boldsymbol{W}^j \prod_{i=l+1}^{L_f} \boldsymbol{W}_{\text{B}}^i\right)^{\top} \widetilde{\boldsymbol{\Sigma}}_{y\boldsymbol{x}_{\text{B}}} \left(\prod_{i=1}^{l-1} \boldsymbol{W}_{\text{B}}^i\right)^{\top}, \ 1 \le l \le L_f, \tag{49}$$

where $\widetilde{\boldsymbol{\Sigma}}_{y\boldsymbol{x}_{\text{B}}} = \boldsymbol{\Sigma}_{y\boldsymbol{x}_{\text{B}}} - \boldsymbol{\Sigma}_{y\boldsymbol{x}_{\text{A}}}\boldsymbol{\Sigma}_{\text{A}}^{-1}\boldsymbol{\Sigma}_{\text{AB}}$. The first-layer weights $\boldsymbol{W}_{\text{B}}^1$ rapidly rotate from $\boldsymbol{\Sigma}_{y\boldsymbol{x}_{\text{B}}}$ to $\widetilde{\boldsymbol{\Sigma}}_{y\boldsymbol{x}_{\text{B}}}$ at time $t_{\text{A}}$ and continues to evolve along $\widetilde{\boldsymbol{\Sigma}}_{y\boldsymbol{x}_{\text{B}}}$ during $t_{\text{A}}$ to $t_{\text{B}}$. Through the same manner of changing variables, we obtain an ordinary differential equation for $u_{\text{B}}$ during $t_{\text{A}}$ to $t_{\text{B}}$:

$$\tau\dot{u}_{\text{B}} = \left\|\boldsymbol{\Sigma}_{y\boldsymbol{x}_{\text{A}}}\boldsymbol{\Sigma}_{\text{A}}^{-1}\right\|^{1-\frac{L_f}{L}} \|\widetilde{\boldsymbol{\Sigma}}_{y\boldsymbol{x}_{\text{B}}}\| u_{\text{B}}^{L_f-1}$$

$$\Rightarrow \quad t - t_{\text{A}} = \tau\left\|\boldsymbol{\Sigma}_{y\boldsymbol{x}_{\text{A}}}\boldsymbol{\Sigma}_{\text{A}}^{-1}\right\|^{\frac{L_f}{L}-1} \|\widetilde{\boldsymbol{\Sigma}}_{y\boldsymbol{x}_{\text{B}}}\|^{-1} \frac{u_{\text{B}}^{2-L_f} - u_{\text{B}}(t_{\text{A}})^{2-L_f}}{2-L_f}, \ t \in (t_{\text{A}}, t_{\text{B}}). \tag{50}$$

Plugging in $u_{\text{B}}(t_{\text{A}})$ obtained in Eq. (48), we get the time $t_{\text{B}}$:

$$t_{\text{B}} = t_{\text{A}} + \tau\frac{u_{\text{B}}(t_{\text{B}})^{2-L_f} - u_{\text{B}}(t_{\text{A}})^{2-L_f}}{(2-L_f)\left\|\boldsymbol{\Sigma}_{y\boldsymbol{x}_{\text{A}}}\boldsymbol{\Sigma}_{\text{A}}^{-1}\right\|^{1-\frac{L_f}{L}} \|\widetilde{\boldsymbol{\Sigma}}_{y\boldsymbol{x}_{\text{B}}}\|}$$

$$\approx t_{\text{A}} + \tau\frac{-u_{\text{B}}(t_{\text{A}})^{2-L_f}}{(2-L_f)\left\|\boldsymbol{\Sigma}_{y\boldsymbol{x}_{\text{A}}}\boldsymbol{\Sigma}_{\text{A}}^{-1}\right\|^{1-\frac{L_f}{L}} \|\widetilde{\boldsymbol{\Sigma}}_{y\boldsymbol{x}_{\text{B}}}\|} \tag{51}$$

$$\approx t_{\text{A}} + \tau\frac{1 - \|\boldsymbol{\Sigma}_{y\boldsymbol{x}_{\text{A}}}\|^{-1}\|\boldsymbol{\Sigma}_{y\boldsymbol{x}_{\text{B}}}\|}{(L_f-2)\left\|\boldsymbol{\Sigma}_{y\boldsymbol{x}_{\text{A}}}\boldsymbol{\Sigma}_{\text{A}}^{-1}\right\|^{1-\frac{L_f}{L}} \|\widetilde{\boldsymbol{\Sigma}}_{y\boldsymbol{x}_{\text{B}}}\|} u_0^{2-L_f}.$$

Dividing Eq. (51) by Eq. (44), we arrive at the time ratio Eq. (11) in the main text. For intermediate and late fusion linear networks $2 < L_f \le L$, the time ratio is

$$\frac{t_{\text{B}}}{t_{\text{A}}} = 1 + \frac{(\|\boldsymbol{\Sigma}_{y\boldsymbol{x}_{\text{A}}}\| - \|\boldsymbol{\Sigma}_{y\boldsymbol{x}_{\text{B}}}\|)u_0^{L-L_f}}{(L_f-2)\left\|\boldsymbol{\Sigma}_{y\boldsymbol{x}_{\text{A}}}\boldsymbol{\Sigma}_{\text{A}}^{-1}\right\|^{1-\frac{L_f}{L}} \|\boldsymbol{\Sigma}_{y\boldsymbol{x}_{\text{B}}} - \boldsymbol{\Sigma}_{y\boldsymbol{x}_{\text{A}}}\boldsymbol{\Sigma}_{\text{A}}^{-1}\boldsymbol{\Sigma}_{\text{AB}}\|} I(L, L_f)^{-1}, \tag{52}$$

where the integral $I(L, L_f)$ has been defined in Eq. (46).

### D.3 TIME RATIO FOR $L_f = 2$

When the fusion layer is the second layer $L_f = 2$, the equality in Eq. (41) takes following form:

$$\frac{du_{\text{A}}}{du_{\text{B}}} = \frac{\|\boldsymbol{\Sigma}_{y\boldsymbol{x}_{\text{A}}}\| u_{\text{A}}}{\|\boldsymbol{\Sigma}_{y\boldsymbol{x}_{\text{B}}}\| u_{\text{B}}} \quad \Rightarrow \quad \frac{\ln u_{\text{A}} - \ln u_0}{\ln u_{\text{B}} - \ln u_0} = \frac{\|\boldsymbol{\Sigma}_{y\boldsymbol{x}_{\text{A}}}\|}{\|\boldsymbol{\Sigma}_{y\boldsymbol{x}_{\text{B}}}\|}. \tag{53}$$

Consequently, fusion at the second layer is a special case with slightly different expressions for the times. We follow the same procedure as Appendix D.2 and obtain the times for $2 = L_f < L$:

$$t_{\text{A}} \approx \tau\frac{u_0^{2-L}}{\|\boldsymbol{\Sigma}_{y\boldsymbol{x}_{\text{A}}}\|} \int_1^\infty x^{-1}\left(x^2 + x^{2k}\right)^{1-\frac{L}{2}} dx, \tag{54a}$$

$$t_{\text{B}} \approx t_{\text{A}} + \tau\frac{1 - \|\boldsymbol{\Sigma}_{y\boldsymbol{x}_{\text{A}}}\|^{-1}\|\boldsymbol{\Sigma}_{y\boldsymbol{x}_{\text{B}}}\|}{\|\boldsymbol{\Sigma}_{y\boldsymbol{x}_{\text{B}}} - \boldsymbol{\Sigma}_{y\boldsymbol{x}_{\text{A}}}\boldsymbol{\Sigma}_{\text{A}}^{-1}\boldsymbol{\Sigma}_{\text{AB}}\|} \left\|\boldsymbol{\Sigma}_{y\boldsymbol{x}_{\text{A}}}\boldsymbol{\Sigma}_{\text{A}}^{-1}\right\|^{\frac{2}{L}-1} \ln\frac{1}{u_0}. \tag{54b}$$

The time ratio is

$$\frac{t_{\mathrm{B}}}{t_{\mathrm{A}}} = 1 + \frac{(\|\mathbf{\Sigma}_{yx_{\mathrm{A}}}\| - \|\mathbf{\Sigma}_{yx_{\mathrm{B}}}\|)u_0^{L-2}\ln\frac{1}{u_0}}{(L_f - 2)\left\|\mathbf{\Sigma}_{yx_{\mathrm{B}}} - \mathbf{\Sigma}_{yx_{\mathrm{A}}}\mathbf{\Sigma}_{\mathrm{A}}^{-1}\mathbf{\Sigma}_{\mathrm{AB}}\right\|\left\|\mathbf{\Sigma}_{yx_{\mathrm{A}}}\mathbf{\Sigma}_{\mathrm{A}}^{-1}\right\|^{1-\frac{2}{L}}}I(L,2)^{-1}, \qquad (55)$$

where the integral is given by

$$I(L,2) = \int_1^\infty x^{-1}\left(x^2 + x^{2k}\right)^{1-\frac{L}{2}} dx. \qquad (56)$$

### D.4 Time Ratio for Unequal Depth

Our time ratio calculations can be carried out for intermediate fusion linear networks with unequal depth between modalities. Consider a intermediate fusion linear network with $L_c$ post-fusion layers, $L_{\mathrm{A}}$ pre-fusion layers for the modality A branch, and $L_{\mathrm{B}}$ pre-fusion layers for the modality B branch. Assuming $L_{\mathrm{A}}, L_{\mathrm{B}} > 2$ and modality A is learned first, the time ratio is

$$\frac{t_{\mathrm{B}}}{t_{\mathrm{A}}} = 1 + \frac{\frac{u_0^{\frac{L_c + L_{\mathrm{A}} - L_{\mathrm{B}}}{L_{\mathrm{B}}-2}}\|\mathbf{\Sigma}_{yx_{\mathrm{A}}}\| - u_0^{\frac{L_c}{L_{\mathrm{A}}-2}}\|\mathbf{\Sigma}_{yx_{\mathrm{B}}}\|}{\left\|\mathbf{\Sigma}_{yx_{\mathrm{A}}}\mathbf{\Sigma}_{\mathrm{A}}^{-1}\right\|^{\frac{L_c}{L_{\mathrm{A}}+L_c}}\left\|\mathbf{\Sigma}_{yx_{\mathrm{B}}} - \mathbf{\Sigma}_{yx_{\mathrm{A}}}\mathbf{\Sigma}_{\mathrm{A}}^{-1}\mathbf{\Sigma}_{\mathrm{AB}}\right\|}}I(L_{\mathrm{A}}, L_{\mathrm{B}}, L_c)^{-1}, \qquad (57)$$

where the integral is given by

$$I(L_{\mathrm{A}}, L_{\mathrm{B}}, L_c) = \int_1^\infty x^{1-L_{\mathrm{A}}}\left[x^2 + \left(\frac{(L_{\mathrm{B}}-2)\|\mathbf{\Sigma}_{yx_{\mathrm{B}}}\|}{(L_{\mathrm{A}}-2)\|\mathbf{\Sigma}_{yx_{\mathrm{A}}}\|}u_0^{L_{\mathrm{B}}-L_{\mathrm{A}}}\left(x^{2-L_{\mathrm{A}}} - 1\right) + 1\right)^{\frac{2}{2-L_{\mathrm{B}}}}\right]^{-\frac{L_c}{2}} dx. \qquad (58)$$

As a sanity check, setting $L_{\mathrm{A}} = L_{\mathrm{B}}$ gives us back the same expression as Eq. (11).

## E Feature Evolution in Multimodal Linear Networks

We compare the feature evolution in two-layer early fusion and late fusion linear networks to gain a more detailed understanding of their different learning dynamics. As studied by Atanasov et al. (2022), features of linear networks lie in the first-layer weights. We thus plot the first-layer weights in multimodal linear networks at different times of training to inspect the feature evolution.

### E.1 Feature Evolution in Early Fusion Linear Networks

In early fusion linear networks with small initialization, the balancing property $\boldsymbol{W}^1\boldsymbol{W}^{1\top} = \boldsymbol{W}^{2\top}\boldsymbol{W}^2$ holds true throughout training, which implies $\boldsymbol{W}^1$ is rank-one throughout training (Du et al., 2018; Ji & Telgarsky, 2019). Specifically, $\boldsymbol{W}^1$ initially aligns with $\mathbf{\Sigma}_{yx}$ during the plateau and eventually aligns with $\mathbf{\Sigma}_{yx}\mathbf{\Sigma}^{-1}$ after $\boldsymbol{W}^1$ have grown in scale and rotated during the brief transitional period. We illustrate this process with Fig. 8b and videos in the supplementary material. In deep early fusion linear networks, $\boldsymbol{W}^1$ behaves qualitatively the same.

### E.2 Feature Evolution in Intermediate and Late Fusion Linear Networks

In intermediate and late fusion linear networks, the balancing property takes a different form as given in Eqs. (34) and (35). Thus the first-layer weights are not constrained to be rank-one. Specifically, $\boldsymbol{W}_{\mathrm{A}}^1$ grows during the first transitional period while $\boldsymbol{W}_{\mathrm{B}}^1$ remains close to small initialization. After a unimodal phase, $\boldsymbol{W}_{\mathrm{B}}^1$ starts to grow during the second transitional period while $\boldsymbol{W}_{\mathrm{A}}^1$ stays unchanged, shrinks in scale, or expands in scale depending on modality B has zero, positive, or negative correlation with modality A. We illustrate this process with Fig. 8d and videos in the supplementary material. In deep late and intermediate fusion linear networks, $\boldsymbol{W}^1$ behaves qualitatively the same as in two-layer late fusion linear networks.

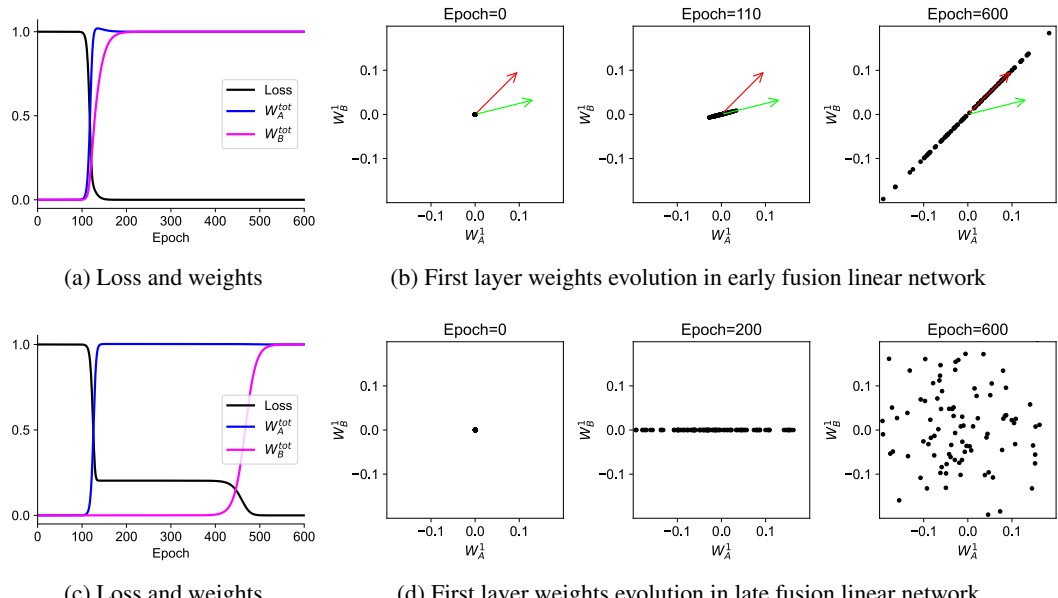

Figure 8: Feature evolution in two-layer early fusion and late fusion linear networks. We plot the feature evolution, which are first-layer weights in linear networks, corresponding to dynamics in Figs. 2b and 2d. (a) Time trajectories of loss and total weights of the same two-layer early fusion network as Fig. 2b. (b) First-layer weights at initialization, during training, and at convergence in (a). The green arrow denotes the direction of $\mathbf{\Sigma}_{y\boldsymbol{x}}$. The red arrow denotes the direction of $\mathbf{\Sigma}_{y\boldsymbol{x}}\mathbf{\Sigma}^{-1}$. (c) Time trajectories of loss and total weights of the same two-layer late fusion network as Fig. 2d. (d) First-layer weights at initialization, during training, and at convergence in (c). Complementary videos can be found in the supplementary material.

## F  SUPERFICIAL MODALITY PREFERENCE

We impose that modality A contributes less to the output, meaning the loss would decrease more if the network visited a saddle in $\mathcal{M}_\mathrm{B}$ instead of $\mathcal{M}_\mathrm{A}$ as illustrated in Fig. 4a:

$$\mathcal{L}(\mathcal{M}_\mathrm{A}) > \mathcal{L}(\mathcal{M}_\mathrm{B}) \tag{59}$$

Substituting in the saddles defined in Eq. (6), we expand and simplify the two mean square losses:

$$
\begin{aligned}
\mathcal{L}(\mathcal{M}_\mathrm{A}) &= \left\langle (y - \mathbf{\Sigma}_{y\boldsymbol{x}_\mathrm{A}}\mathbf{\Sigma}_\mathrm{A}^{-1}\boldsymbol{x}_\mathrm{A})^2 \right\rangle \\
&= \left\langle y^2 \right\rangle - 2\mathbf{\Sigma}_{y\boldsymbol{x}_\mathrm{A}}\mathbf{\Sigma}_\mathrm{A}^{-1}\langle y\boldsymbol{x}_\mathrm{A}\rangle + \mathbf{\Sigma}_{y\boldsymbol{x}_\mathrm{A}}\mathbf{\Sigma}_\mathrm{A}^{-1}\left\langle \boldsymbol{x}_\mathrm{A}\boldsymbol{x}_\mathrm{A}^\top \right\rangle \mathbf{\Sigma}_\mathrm{A}^{-1}\mathbf{\Sigma}_{y\boldsymbol{x}_\mathrm{A}}^\top \\
&= \left\langle y^2 \right\rangle - \mathbf{\Sigma}_{y\boldsymbol{x}_\mathrm{A}}\mathbf{\Sigma}_\mathrm{A}^{-1}\mathbf{\Sigma}_{y\boldsymbol{x}_\mathrm{A}}^\top, 
\end{aligned} \tag{60a}
$$

$$
\begin{aligned}
\mathcal{L}(\mathcal{M}_\mathrm{B}) &= \left\langle (y - \mathbf{\Sigma}_{y\boldsymbol{x}_\mathrm{B}}\mathbf{\Sigma}_\mathrm{B}^{-1}\boldsymbol{x}_\mathrm{B})^2 \right\rangle \\
&= \left\langle y^2 \right\rangle - 2\mathbf{\Sigma}_{y\boldsymbol{x}_\mathrm{B}}\mathbf{\Sigma}_\mathrm{A}^{-1}\langle y\boldsymbol{x}_\mathrm{B}\rangle + \mathbf{\Sigma}_{y\boldsymbol{x}_\mathrm{B}}\mathbf{\Sigma}_\mathrm{B}^{-1}\left\langle \boldsymbol{x}_\mathrm{B}\boldsymbol{x}_\mathrm{B}^\top \right\rangle \mathbf{\Sigma}_\mathrm{B}^{-1}\mathbf{\Sigma}_{y\boldsymbol{x}_\mathrm{B}}^\top \\
&= \left\langle y^2 \right\rangle - \mathbf{\Sigma}_{y\boldsymbol{x}_\mathrm{B}}\mathbf{\Sigma}_\mathrm{B}^{-1}\mathbf{\Sigma}_{y\boldsymbol{x}_\mathrm{B}}^\top. 
\end{aligned} \tag{60b}
$$

Plugging Eq. (60) into Eq. (59) gives us

$$\mathbf{\Sigma}_{y\boldsymbol{x}_\mathrm{A}}\mathbf{\Sigma}_\mathrm{A}^{-1}\mathbf{\Sigma}_{y\boldsymbol{x}_\mathrm{A}}^\top < \mathbf{\Sigma}_{y\boldsymbol{x}_\mathrm{B}}\mathbf{\Sigma}_\mathrm{B}^{-1}\mathbf{\Sigma}_{y\boldsymbol{x}_\mathrm{B}}^\top. \tag{61}$$

Since we are also assuming modality A is learned first, which is true when $\|\mathbf{\Sigma}_{y\boldsymbol{x}_\mathrm{A}}\| > \|\mathbf{\Sigma}_{y\boldsymbol{x}_\mathrm{B}}\|$. We thereby arrive at the two inequality conditions Eq. (9) in the main text.

## G  UNDERPARAMETERIZATION AND OVERPARAMETERIZATION IN MULTIMODAL LINEAR NETWORKS

We have studied multimodal networks trained with sufficient noiseless data so far. In this section, we consider the case where the training set contains noise and the amount of training samples vary.

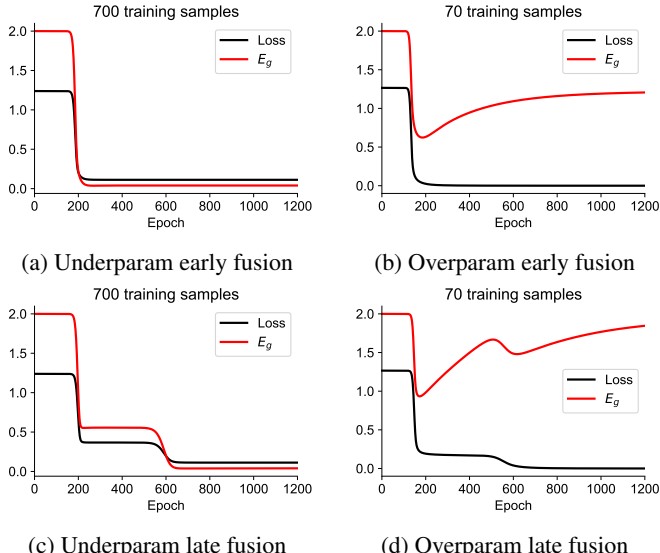

(a) Underparam early fusion      (b) Overparam early fusion

(c) Underparam late fusion      (d) Overparam late fusion

Figure 9: Overparameterized and underparameterized two-layer early and late fusion linear networks. Both modalities have 50 input dimensions with a total of 100. Further imlpementation details can be found in Appendix I. The input covariance is $\Sigma_A = I, \Sigma_B = 3I, \Sigma_{AB} = 0$. The target output is generated as $y = w^* x + \epsilon$, where $w^* = 1/10$ and the noise is independently sampled from $\epsilon \sim \mathcal{N}(0, 0.5^2)$. (a,b) Loss and generalization error trajectories of a two-layer early fusion linear network trained with 700, 70 training samples. (c,d) Loss and generalization error trajectories of a two-layer late fusion linear network trained with 700, 70 training samples.

## G.1 UNDERPARAMETERIZED REGIME

Training loss well reflects the corresponding generalization error in the underparameterized regime as shown in Figs. 9a and 9c, where there is sufficient amount of data and thus the training data distribution is close to the true data distribution. The convergent generalization error is lower than the convergent training loss because the training set contains noise. In the underparameterized regime, analysis on the training loss translates to the generalization error.

## G.2 OVERPARAMETERIZED REGIME

In the overparameterized regime, the number of samples is insufficient compared to the number of effective parameters, which is the input dimensions for linear networks (Advani et al., 2020).

As shown in Fig. 9b, the overparameterized early fusion linear network learns both modalities during one transitional period. The generalization error decreases during the transitional period and increases afterwards as predicted by theory (Advani et al., 2020). If early stopping is adopted, we obtain a model that has learned from both modalities and not overfitted.

As shown in Fig. 9d, the overparameterized late fusion linear network learns the faster-to-learn modality first and overfits this modality during the unimodal phase when the training loss plateaus but the generalization error increases. In this case, there is a dilemma between overfitting one modality and underfitting the other. If optimal early stopping on the generalization error is adopted, training terminates right after the first modality is learned while the second modality has not, yielding a model with strong and permanent unimodal bias. If training terminates after both modalities are learned, the network has overfitted the first modality, which can yield a generalization error higher than that in early stopping. Thus overfitting is a mechanism that can convert the transient unimodal phase to a generalization deficit and permanent unimodal bias.

As identified with empirical evaluations in Wang et al. (2020), different modalities overfit and generalize at different speeds in multimodal networks. With theoretical evidence, we elaborate on that in late fusion linear network, the difference in speeds gives rise to the unimodal phase in both un-

derparameterized and overparameterized regime and further causes the dilemma between overfitting one modality and underfitting the other in the overparameterized regime.

# H TWO-LAYER RELU NETWORKS

## H.1 RELU NETWORKS WITH LINEAR TASK

Two-layer ReLU networks are trained to learn the same linear task introduced in Fig. 2. The loss and weights trajectories in Fig. 10 are qualitatively the same as Fig. 2, except that learning evolves about two times slower and the converged weights are two times larger.

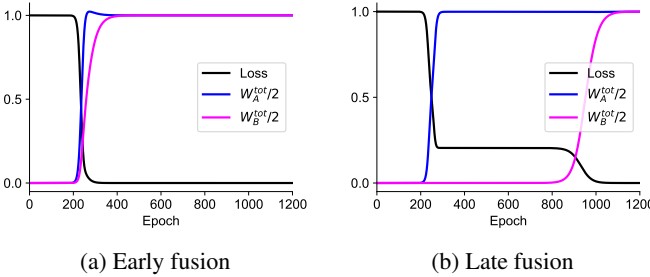

(a) Early fusion                    (b) Late fusion

Figure 10: Two-layer early fusion and late fusion ReLU networks trained on a linear tasks. The same dataset, number of neurons, and initialization scale as Fig. 2 are used, except that ReLU activation is added to the hidden layer. (a,b) Time trajectories of loss and total weights in the two-layer early fusion (a) and late fusion (b) ReLU network.

## H.2 RELU NETWORKS WITH NONLINEAR TASK

We consider a simple nonlinear task of learning $y = x_A + \text{XOR}(x_B)$, where $x_A \in \mathbb{R}, x_B \in \{[1,1],[1,-1],[-1,1],[-1,-1]\}$. The term $\text{XOR}(x_B)$ refers to performing exclusive-or to the two dimensions of $x_B$. We train two-layer early fusion and late fusion ReLU networks to learn this task with variance of the linear modality $\text{Var}(x_A) \equiv \sigma_A = 1, 2$ or 3. We inspect the features in ReLU networks by plotting the first-layer weights since the features in a two-layer ReLU network lie in its first-layer weights (Xie et al., 2017).

As shown in Figs. 11b, 11d and 11f, two-layer late fusion ReLU networks always solve the task by consistently forming the four perpendicular XOR features. We can see two transitions in the loss trajectories of late fusion ReLU networks, which are similar to the transitions in linear networks. Late fusion ReLU newtorks learn the XOR modality during one transitional period and learn the linear modality during the other. As an example of converged features shown in Fig. 11d, $W_B^1$ has taken on the rank-two XOR structure and $W_A^1$ has grown in scale while preserving its independence from $W_B^1$ at initialization.

As shown in Figs. 11a, 11c and 11e, two-layer early fusion ReLU networks struggle to extract the XOR features. In the early stage of training, features in $W^1$ favor particular directions in the three-dimensional space that are most correlated with the target output in early fusion ReLU networks. In comparison with features in late fusion ReLU networks, the first layer weights for modality A do not preserve its independence from weights modality B as shown in Fig. 11c. The larger the variance of the linear modality, the closer the favorable direction is to the direction of the linear modality. In later stage of training, features in $W^1$ can rotate or scatter, giving rise to multiple transitional periods as shown in Fig. 11c and its corresponding video in the supplementary material. For a large variance in the linear modality, the features are highly aligned with the linear modality direction and the network can be stuck in a local minimum, failing to learn from the XOR modality as in Fig. 11e.

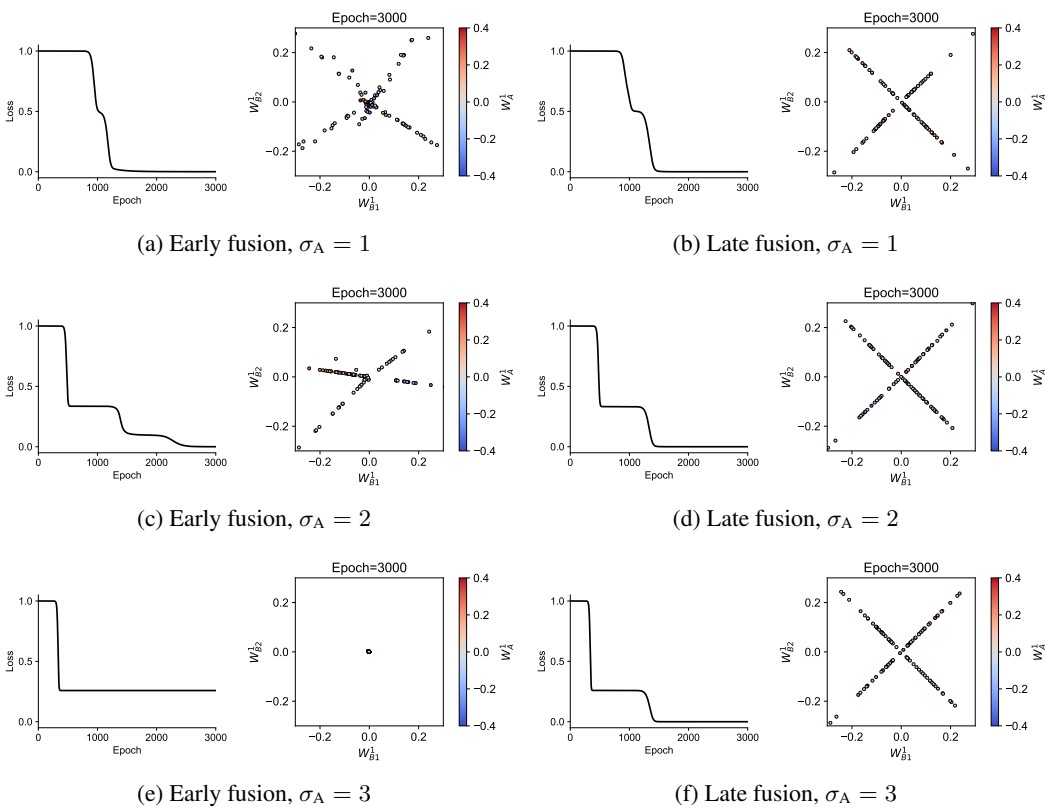

Figure 11: Two-layer early fusion and late fusion ReLU networks trained on a XOR and linear task. The early fusion network has 100 hidden neurons. The late fusion network has 100 hidden neurons in both branches. In every panel, the loss trajectory is plotted on the left and the first-layer weights at the end of training are plotted on the right. The first-layer weights are three-dimensional: the two dimensions of $\boldsymbol{W}_{\mathrm{B}}^1$ are plotted as the position of the dots and $\boldsymbol{W}_{\mathrm{A}}^1$ is plotted as the color of the dots. (a,c,e) Two-layer early fusion ReLU networks. (b,d,f) Two-layer late fusion ReLU networks. Complementary videos can be found in the supplementary material.

# I IMPLEMENTATION DETAILS

## I.1 DATA GENERATION

For data generation, we sample input data points from zero mean normal distribution $\boldsymbol{x} \sim \mathcal{N}(\boldsymbol{0}, \boldsymbol{\Sigma})$ and generate the corresponding target output from a groundtruth linear map $y = \boldsymbol{w}^* \boldsymbol{x}$. All datasets do not contain noise except Fig. 9. The number of training samples is 4096 for all experiments except Fig. 9. Hence, all experiments, except Fig. 9, fall into the underparameterized regime where the training loss well reflects the generalization error.

Note that we do not lose generality by using linear datasets because the learning dyanmics of linear networks as given in Eqs. (3) and (4) only concern the input correlation matrix $\boldsymbol{\Sigma}$ and input-output correlation matrix $\boldsymbol{\Sigma}_{y\boldsymbol{x}}$. Hence, datasets generated from any distribution and target map with the same $\boldsymbol{\Sigma}, \boldsymbol{\Sigma}_{y\boldsymbol{x}}$ will have the same learning dynamics in linear networks.

## I.2 MULTIMODAL LINEAR NETWORKS

All early fusion networks we used have 100 neurons in every hidden layer. Late fusion networks have 100 neurons in every hidden layer for both branches. Intermediate fusion networks have 100 neurons in every pre-fusion hidden layer for both branches and 200 neurons in every post-fusion hidden layer. All networks are trained with full batch gradient descent with learning rate $\eta = 0.04$. For Figs. 3b, 3c and 5b to 5d, we run experiments with 5 random seeds and report the average.

