# OpenReview forum: "A Theory of Unimodal Bias in Multimodal Learning"
_ICLR.cc/2024/Conference — Submitted to ICLR 2024_

### Official Review · Reviewer_T6g2 · 2023-10-12

**Soundness:** 2 fair
**Presentation:** 1 poor
**Contribution:** 2 fair
**Rating:** 3
**Confidence:** 4

**Summary:**

This work initiates a theoretical study on the unimodal bias in multimodal learning, by analyzing the training dynamics. In particular, for linear networks, factors including a deeper fusion layer, stronger correlations between modalities and disparities in input-output correlations are identified as the causes of the unimodal bias.

**Strengths:**

The problem of building up a theoretical understanding of multimodal learning is urgent and significant.

Analyzing the training dynamic is an interesting and promising avenue for understanding unimodal bias. Such direction is intuitive due to relevant works on the implicit bias of neural networks, particularly on the training dynamics after zero training loss.

**Weaknesses:**

**Unfocused writing:** the writing of this work is unfocused. From the title and the contents, I presume this work is theoretical paper. However, there is no formal presentation of the theoretical results (propositions/lemmas/theorems). It greatly obstructs a smooth understanding of the results for readers. After I read the main-text, I still can't tell which part is prioritized. There should be rigorous summaries of the theoretical results. Even a heuristic-level summary can be useful.

**Restricted results:** the dynamic analysis only concerns linear networks, which is not popular in practice. Tools from the study of implicit bias of neural networks (for example, [1]) might be useful to deal with nonlinear networks.

[1] What Happens after SGD Reaches Zero Loss? – A Mathematical Framework, Li et al 2021

**Questions:**

Though the contents might be interesting, the current writing style is certainly non-standard and disadvantageous to readability. In my opinion, the work needs a thorough rewriting for a clear summary and presentation of the theoretical results.

Still, I'm curious the reason of the current writing. There seems to be plenty rigorous mathematical arguments in the appendix. Is there any difficulty preventing you from summarizing them as theorems?

---

> ### Author Response · Authors · 2023-11-18
> **Response to Reviewer T6g2**
>
> We thank the reviewer for the questions and suggestions and would like to respond below.
>
> - Restricted results
>
>   We agree that our analytical analysis is on the learning dynamics of linear networks. Nonetheless, we empirically find that our results, derived for linear networks, carry over to two-layer early/late fusion ReLU networks when the target task is linear. The results of two-layer ReLU networks (plotted as crosses in Fig. 3b and 3c) closely follow the theoretical predictions and results of their linear counterparts.
>
>   Furthermore, we believe understanding behaviors of multimodal deep linear networks is a prerequisite to tackling multimodal nonlinear networks. Studying linear networks has restrictions in the network architecture, while studying nonlinear networks has its restrictions as well --- the amount of amenable questions in nonlinear networks are much fewer. Focusing on linear networks grants us opportunities to derive results that would be highly intractable in nonlinear networks. Thus our novelty lies in that we derive the time ratio of when two modalities get learned in terms of dataset statistics, network configuration, and initialization. This was never done analytically for any multimodal networks and is still intractable for nonlinear networks to the best of our knowledge.
>
> - Unfocused writing
>
>   Regarding our writing, we intended to strike a balance between formality and accessibility to serve the primal goal of letting as many readers understand our content as possible. We note that the other three reviewers explicitly mentioned our manuscript is well-written and clear as a strength. Propositions/lemmas/theorems is one of the mainstream writing styles for theory but not the only style. For instance, the following top conference/journal papers [1,2,3,4] did not write propositions/lemmas/theorems but did a great job in delivering theoretical contents. We have used style conventions common in the statistical physics of machine learning community.
>
>   Regarding our priorities, we would like to restate them here. We studied the duration of the unimodal phase, misattribution during the unimodal phase, and superficial modality preference in this paper. Among the three topics, the duration of the unimodal phase is prioritized. Our main theoretical result is the time ratio in Eq. (11-12), from which we learn a deeper fusion layer, a larger input-output correlation ratio, and stronger correlations between input modalities prolong the unimodal phase in the joint training of deep multimodal linear networks with small initialization. Should you find specific sections unclear or incoherent, please let us know. We will try our best to improve the clarity.
>
>   [1] Atanasov, et al. "Neural Networks as Kernel Learners: The Silent Alignment Effect." ICLR. 2022.
>
>   [2] Canatar, et al. "Spectral bias and task-model alignment explain generalization in kernel regression and infinitely wide neural networks." Nature communications. 2021.
>
>   [3] Gerace, et al. "Generalisation error in learning with random features and the hidden manifold model." ICML. 2020.
>
>   [4] Saxe, et al. "Exact solutions to the nonlinear dynamics of learning in deep linear neural networks." ICLR. 2014

---

> > ### Comment · Reviewer_T6g2 · 2023-11-22
> >
> > Thanks for the reply. I agree with you that writing theorems is not the only writing style. However, it's the mainstream style for theory, which means it's the most acceptable style for readers in the ICLR community.
> >
> > Since the criteria of accepting a paper also concerns how it fits the interests of the community, it's important to make most readers read smoothly. It's likely the current style is standard for statistical physics of machine learning, but I'm not familiar with statistical physics just as most of the ICLR community. The current writing makes it hard for me to judge the results, and I sincerely suggest the authors to consider changing the writing style, if you are submitting to mainstream ML conferences.

---

### Official Review · Reviewer_H94d · 2023-10-27

**Soundness:** 3 good
**Presentation:** 3 good
**Contribution:** 2 fair
**Rating:** 5
**Confidence:** 4

**Summary:**

This paper theoretically study the unimodal bias in deep multimodal linear networks. They also derived the duration of the unimodal phase in terms of network configuration, and dataset statistics. The theoretical findings are supported by numerical simulations.

**Strengths:**

1. The paper is clear and well-written
2. The results for early fusion and intermediate fusion are novel
3.  This paper explicitly characterizes an analytical relationship between unimodal bias, network configuration, and dataset statistics under simplified linear settings, and the implication from theory, fast-to-learn modality, is interesting.
4.  The results are validated by numerical simulations.

**Weaknesses:**

1. The author claims "there is a scarce theoretical understanding of how unimodal bias arises and how it is affected by the network configuration, dataset statistics, and initialization". However,  the work [1] mentioned in this paper already theoretically explored the rise of unimodal bias in a more realistic neural network setting, and their results somewhat reveal the relationship between the inferior performance of late-fusion networks with initialization and modality correlations.

2. Moreover, there is another work [2] that has provided some analysis about insufficient learning of uni-modal features and proposed some methods to overcome the limitations of late-fusion networks.  In their study, they also discuss the effect of easy-to-learn features.

Therefore, the novelty and contribution of this paper compared to the previous analysis is not clear to me, considering they studied more complex and realistic settings.

[1] Modality Competition: What Makes Joint Training of Multi-modal Network Fail in Deep Learning? (Provably),  Huang et al, ICML 2022

[2] On Uni-Modal Feature Learning in Supervised Multi-Modal Learning, Du et al, ICML 2023

**Questions:**

See weakness.

1. For the data generation process,  it appears that only the correlation matrices are required. Are there specific assumptions made regarding $y$ that need to be clarified?
2. The author claims "we develop a theory of unimodal bias with deep multimodal linear networks." However, the frequent use of the approximation symbol ($\approx$) in the appendix when deriving mean results raises questions about the rigor and precision of the theoretical justifications provided.

---

> ### Author Response · Authors · 2023-11-18
> **Response to Reviewer H94d**
>
> ## Comparison with Two Prior Works
>
> We thank the reviewer for highlighting the two related papers and would like to provide a more detailed comparison between our work and theirs.
>
> - Huang et al studied two-layer late fusion network with smooth ReLU activation. We compare the differences between their and our problem setup in the table below. We make a stronger assumption on the activation function. However, in return, we are able to study deep networks with all three common fusion schemes trained with standard gradient descent, all of which were not covered in Huang et al.
>
>   Actually, a main motivation of our work came from the Limitations section in Huang et al, where they point out in their Limitations section that their analysis focused on two-layer late fusion networks and immediate future directions are to study **other fusion frameworks** and **deep neural networks**.
>
>   |                     | Huang et al                                                  | Ours                                     |
>   | ------------------- | ------------------------------------------------------------ | ---------------------------------------- |
>   | Network Depth       | Two-layer                                                    | **Deep**                                 |
>   | Activation function | **Smoothed ReLU**                                            | Linear                                   |
>   | Fusion scheme       | Late fusion                                                  | **Late, intermediate, and early fusion** |
>   | Training            | Fixing the second layer, gradient descent on only the first layer | **Standard gradient descent**            |
>
>   In terms of analytical results, Huang et al proved the occurrence of modality competition. We provide new insights into the process of this competition by studying the learning dynamics. We also derive an analytical expression to describe the influence of dataset statistics and network configurations on unimodal bias, for which Huang et al and other prior literature did not have theory-grounded answers.
>
> - Du et al proposed a solution to mitigate unimodal bias and validated its effectiveness with experiments, which are indeed not covered by our work due to a different scope. On the theory side, Du et al did not analytically study concrete multimodal neural networks. Instead, Du et al developed their theoretical argument by assuming that a multimodal task involves unimodal/paired features and their model learns features in descending order of predicting probability. Specifically, Du et al assumed that powerful features are learned first by citing 2 relevant papers in their footnote 5 on page 5. However, Du et al did not specify how to identify powerful features in a given dataset or why multimodal networks learn powerful features first. Since our work does not assume the dataset comes from a particular data generation process and we study concrete deep multimodal networks with various fusion scheme, we provide novel theory-grounded results that aim to complement the contributions from Du et al.
>
>   Thanks in particular for bringing up this recent paper, we have now added it to our references.
>
> In summary, we believe our paper is not subsumed by the related work, and vice versa. We developed different analytical tools and unraveled different aspects of unimodal bias, which advance our understanding of unimodal bias altogether as a community.
>
> ## Response to Questions
>
> We also appreciate and would like to respond to the two questions raised.
>
> - Thank you for this note. We have added in our main text that we assume the input covariance $\boldsymbol \Sigma$ has full rank but make no further assumption on the correlation matrices.
>
>   Since the gradient descent dynamics of multimodal linear networks given in Eq. (3-4) only involve the correlation matrices $\boldsymbol \Sigma$ and $\boldsymbol \Sigma_{yx}$, knowing $\boldsymbol \Sigma$ and $\boldsymbol \Sigma_{yx}$ suffices to determine the learning trajectories. We have also added an Implementation Details section at the end of Appendix.
>
> - We acknowledge our frequent usage of $\approx$ in the Appendix. To control approximations more explicitly, we are re-working through our derivations to include error correction terms and will post these updates as soon as we are done.
>
>   All our approximation signs are precisely equality when the initialization scale is taken to the limit of $0$. The approximations are also justified by our simulation results, which align well with theoretical predictions.

---

> > ### Comment · Reviewer_H94d · 2023-11-22
> > **Response to Rebuttal**
> >
> > Thanks for the response.  Although the author compared the paper with existing analyses from various aspects, I still found the settings too simplistic and limited to the linear regime. Considering the weaknesses other reviewers pointed out, I have decided to keep my original score.

---

### Official Review · Reviewer_8NBJ · 2023-10-31

**Soundness:** 2 fair
**Presentation:** 3 good
**Contribution:** 2 fair
**Rating:** 5
**Confidence:** 4

**Summary:**

The paper focuses on a theoretical understanding of unimodal bia and  examines the effect of network architecture, dataset characteristics, and initialization factors. It reveals that while early fusion networks do not exhibit unimodal bias, this bias is noticeable in networks with intermediate and late fusion. Additionally, the paper quantifies the duration of the unimodal phase in these settings. To support these findings, the paper presents experimental data using numerical simulations conducted on two-layer ReLU networks and deep linear networks.

**Strengths:**

- The paper tackles an important problem of unimodal bias by investigating the unimodal bias theoretically and understanding the impact of various components such as network configuration, dataset statistics and initialization, which would be of interest to the community.
- The paper was clear and well-written.
- The supporting experimental evidence provides interesting insights into intermediate and late fusion for multimodal learning.

**Weaknesses:**

- The paper presents an interesting study of unimodal bias in intermediate and late fusion contexts, yet the evidence supporting the absence of unimodal bias in early fusion remains unconvincing. The reliance on the Frobenius norm of weights as a metric for understanding unimodal bias seems reasonable, but the experiments regarding early fusion are limited to simplistic scenarios. These toy settings are insufficient to assert the absence of unimodal bias in early fusion.
- All experiments are conducted with linearly separable data. The inclusion of experiments with XOR data, where early fusion fails to perform effectively, further casts doubt on the claims for early fusion. Considering that the main focus of the paper was to investigate the interplay between unimodal bias, network configuration, and dataset statistics, the scope and depth of the study become critical. When this research is contrasted with prior studies that have identified unimodal bias across a diverse range of datasets, the robustness of the current findings for complex, real-world scenarios appears uncertain.

- Minor suggestions
   - In equation 2, did you intend to use W^{tot} alongside y?.
   - The latter part of the caption for figure 2, particularly the description following parts a-c, is somewhat confusing and could benefit from a more straightforward explanation.

**Questions:**

Apart from the review, I have some additional questions:
- Can the authors provide more details with respect to Fig 3e)
- “Two-layer early fusion ReLU networks do not learn XOR features and can even fail to learn this task” – can the authors comment and provide more reasoning about this observation wirth respect to the XoR experiment?

---

> ### Author Response · Authors · 2023-11-18
> **Response to Reviewer 8NBJ (part 1)**
>
> We thank the reviewer for the questions and suggestions and would like to respond below.
>
> - Absence of unimodal bias in early fusion
>
>   We would like to clarify that we only claim that unimodal bias is absent in early fusion linear networks. And we strongly agree that we cannot claim the same for other nonlinear early fusion networks. For early fusion linear networks, we studied linearly separable data without losing generality since linear networks can only learn the linear component even if the dataset contains a nonlinear task. In addition to the one example in Fig 2, we also numerically computed the time ratio for four-layer early fusion linear networks in Fig 5b-d (purple circles), which are all close to 1.
>
>   It is also important to note that the learning curves for different modalities are not exactly the same in early fusion. By absence, we intend to mean that we see no conspicuous unimodal phase during training. Since early fusion linear networks learn both modalities during one transitional period, the time lag is bounded by the duration of the transitional period, which approaches zero when the initialization scale is taken to the limit of $0$. In comparison, the time lag is much longer in late fusion linear networks since two modalities are learned during two separated transitional periods with a unimodal phase in between. We can revise the wording "absent" if it is the word that particularly misleads readers.
>
> - Uncertain robustness for real-world scenarios
>
>   Our results provide a fundamental understanding of unimodal bias in deep linear networks, and in this specific setting, earlier fusion does shorten the unimodal phase. However, this analysis is only one part of a more complete picture: in nonlinear settings, several pre-fusion layers may be necessary to implement nonlinear transformations. This is why we include experiments on XOR, in order to show the additional factors that enter in nonlinear settings. We hope our work provides practitioners with some theoretical intuitions and theorists with a prerequisite to study unimodal bias in more complex scenarios.
>
>   We note that all theoretical approaches at present study simplified settings to obtain greater tractability, and this work is one step toward a more complete treatment. Our intuition and qualitative conclusions can still apply to complex scenarios. For instance, the idea that a shallower fusion layer enables modalities to cooperate more reciprocally and alleviates unimodal bias still holds in complex multimodal networks. This is because it remains true that the gradient descent updates for each layer share more common terms when the fusion layer is shallower.
>
>   We have also found our results consistent with empirical findings while containing more details for our specific setup. To give examples, multimodal networks were found to utilize different modalities at different speeds during training in [1,2], which is consistent with our time ratio results. [3] found that balancing the dataset mitigates unimodal bias, which is consistent with our conclusion that smaller disparities in input-output correlations reduce unimodal bias.
>
>   [1] Wu, et al. "Characterizing and overcoming the greedy nature of learning in multi-modal deep neural networks." ICML. 2022.
>
>   [2] Wang, et al. "What makes training multi-modal classification networks hard?." CVPR. 2020.
>
>   [3] Goyal, et al. "Making the v in vqa matter: Elevating the role of image understanding in visual question answering." CVPR. 2017.
>
> - More comments on the XOR experiment
>
>   We thank the reviewer for their interest in our XOR experiment. We have added videos in the supplementary material and extended the Appendix H.2 on the XOR experiment. The videos show the time evolution of features corresponding to Fig 11.
>
>   Late fusion ReLU networks learn the XOR and linear task better because the XOR branch of the network forms the XOR features in the two-dimensional space without being interfered by the linear branch. The features in the XOR branch preserve its independence from the linear branch as how the weights are independently initialized. However, early fusion ReLU networks favor particular feature directions in the three-dimensional space that correlate the most with the target output. The favorable feature directions tilt more towards the direction of the linear modality and are more prone to let the network be stuck in local minimum when the variance in the linear modality is larger.

---

> ### Author Response · Authors · 2023-11-18
> **Response to Reviewer 8NBJ (part 2: minor clarifications)**
>
> ## Minor clarifications
>
> - Notation in Eq 2
>
>   We checked that we do intend to write down $\boldsymbol W^{\text{tot}}{\boldsymbol x}$. We re-write Eq 2 with more details below:
>   $$
>   \hat y (\boldsymbol x; \boldsymbol W) = \prod _{j=L_f+1}^L \boldsymbol W^j \left( \prod _{i=1}^{L_f} \boldsymbol W^i_A \boldsymbol x_A + \prod _{i=1}^{L_f} \boldsymbol W^i_B \boldsymbol x_B \right)
>   \equiv \boldsymbol W^{\text{tot}}_A \boldsymbol x_A + \boldsymbol W^{\text{tot}}_B \boldsymbol x_B
>   \equiv \boldsymbol W^{\text{tot}} \boldsymbol x  ,
>   $$
>   where
>   $$
>   \boldsymbol W^{\text{tot}}_A \equiv  \prod _{j=L_f+1}^L \boldsymbol W^j \prod _{i=1}^{L_f} \boldsymbol W^i_A ,
>   $$
>
>   $$
>   \boldsymbol W^{\text{tot}}_B \equiv \prod _{j=L_f+1}^L \boldsymbol W^j \prod _{i=1}^{L_f} \boldsymbol W^i_B  ,
>   $$
>
>   $$
>   \boldsymbol W^{\text{tot}} \equiv \left[ \boldsymbol W^{\text{tot}}_A, \boldsymbol W^{\text{tot}}_B \right] .
>   $$
>
> - Caption of Fig 2
>
>   We apologize for inducing confusion and have changed the order of the caption for Fig 2. In Fig 2b and 2d, we plot the loss and total weight trajectories with respect to time (i.e., epoch) for a 2D simple case. The loss trajectory is the training mean square error loss. The total weights are $\boldsymbol W^{\text{tot}}_A, \boldsymbol W^{\text{tot}}_B$ as defined in Eq 2 and clarified above. Since we are assuming scalars $x_A, x_B$ in Fig 2, $\boldsymbol W^{\text{tot}}_A, \boldsymbol W^{\text{tot}}_B$ are also scalars and thus can be directly plotted.
>
>   Please let us know if this is still unclear or we got the question wrong.
>
> - Caption of Fig 3e
>
>   We realize it can be confusing that we plotted the time ratio twice in Fig 3b and 3e and want to re-try explaining.
>
>   Lines in Fig 3b and the entire Fig 3e are plotted with theoretical predictions. Specifically, the theoretical prediction here is a 2D corollary to the general time ratio in late fusion linear networks given in Eq 8. By plugging the standard 2D covariance matrix $\boldsymbol\Sigma = \left[ \sigma_A^2 , \rho \sigma_A \sigma_B ; \rho \sigma_B \sigma_A , \sigma_B^2 \right]$ and target output $y = x_A + x_B$ into the Eq 8, we obtain the time ratio in the special case of 2D inputs:
>   $$
>   \frac{t_B}{t_A} = 1 + \frac{\frac{\sigma_A^2}{\sigma_B^2}-1}{1-\rho^2} .
>   $$
>   We plot Fig 3b to show that our theoretical predictions align closely with simulations. We plot Fig 3e to show how the theoretical predictions vary with continuous values of correlation $\rho$ and variance ratio $\sigma_A/\sigma_B$. We are open to the removal Fig 3e if most readers find it redundant.
>
>   The rationale for assuming 2D inputs in Fig 3 is that metrics for variances inside a modality and correlations across modalities are well-defined in 2x2 covariance matrices, which allows us to systematically sweep the variance and correlation parameters. Furthermore, we have found that covariance matrices with higher dimensions do not give rise to any qualitatively different behavior as long as the input dimension is not larger than the number of training samples.

---

### Official Review · Reviewer_W8DV · 2023-11-01

**Soundness:** 3 good
**Presentation:** 3 good
**Contribution:** 2 fair
**Rating:** 8
**Confidence:** 3

**Summary:**

The paper studies how deep linear networks with multiple pathways learn from data to produce a scalar output  when starting from small weights. The paper studies how the learning dynamics depend on layer in the network architechture when the modalities is fused in an additive manner, and find that with early fusion both modalities are learned (approx) simultaenously, whereas with late fusion the modality more correlated with the output is learned earlier in training.

**Strengths:**

The paper is well-written, the figures are clear, and the mathematical results appear to be sound. I did not carefully check the derivation of the time ratios for learning from the different modalities in the Appendix.

**Weaknesses:**

The motivating phenomonom (unimodal bias) that one modality dominates at convergence is not addressed in the deep linear multimodal settings, as the manuscript studies the transient dynamics for when these modalities get learned (which the authors directly acknowledge in the intro). While the motivating phenomenon is well motivated, developing an improved understanding of the transient dynamics was not well-motivated. It would be helpful if the manuscript could comment/discuss how the analysis of the transient in the deep linear network setting could inform the phenomemon of unimodal bias at convergence in practice. I also feel that the paper title could better reflect the contents of the paper.

Do the results extend to the multitask case where output y is a vector?

**Questions:**

The authors considered architectures of equivalent depth between pathways. How do the result change if these depths differ?

How do the weights evolve in the pre and post fusion layers?

When the paper says: "In essence, an early fusion point allows the weaker modality to benefit from the stronger modality's learning in the post-fusion layers:" Are there settings where this can be harmful as well, or would the larger scale of the weights always help learning?

Minor:

What matrix norm is being used throughout the paper? It should be clarified (For example in Eq 7, Eq 9 etc). Apologies if I missed it.
Do the results apply to more complicated covariance matrices? It seems like diagonal input covarainces were studied, and 2x2 matrices.

Define the product notation used for a product over weight matrices (for example in Eq 2)

It was unclear the experimental details used. For example, were there a finite amount of inputs used, or were inputs drawn according to the covariance structure every batch (In Fig 2,3; for example). The paper mentioned full-batch SGD but the details were not provided (and could not find in appendix.)

In sect. 3.2.3 unclear why it is ideal for modality learned first to lead to larger decrease in loss.

Wording " a smaller initialization scale exacerbates the impediment to learning modality A compared to modality
B, yielding a larger time ratio" is unclear.

It was a bit strange to add in new results in the discussion section.

Why do the results require a small initialization?

---

> ### Author Response · Authors · 2023-11-18
> **Response to Reviewer W8DV (part 1)**
>
> We appreciate your reviewing our manuscript in great depth and detail. We have taken your constructive suggestions and updated our manuscript. Additionally, we respond to the questions and highlight our edits below.
>
> - Motivation of studying transient unimodal bias
>
>   Thank you for the important comment, we believe we in fact do study a situation that is relevant to the non-transient setting, and have added experiments to demonstrate why. In particular, we note that in practical settings, networks can overfit such that their training error differs from test error. As shown in new experiments in the revision Fig 9d, an overparameterized late fusion linear network learns the faster-to-learn modality first and overfits this modality during the unimodal phase when the training loss plateaus but the generalization error increases. In this case, there is a dilemma between overfitting one modality and underfitting the other. Under optimal early stopping on the generalization error, training terminates right after the first modality is learned, and the network has not learned the second modality, yielding a model with a strong and **permanent** unimodal bias. If training terminates after both modalities are learned, the network has overfitted the first modality, which can yield a generalization error higher than that in early stopping. Thus overfitting is a mechanism that can convert the transient unimodal phase to a generalization deficit and permanent unimodal bias.
>
>   An additional note is that even in underparameterized networks, the transient unimodal phase will lead to a bias at convergence when input modalities have collinearity. In the collinear input case, late and intermediate fusion networks learn to fit the output with only the faster-to-learn modality and never learn the rest. Such multimodal networks are unable to make predictions in the absence of certain modalities, which is a common task in multimodal learning.
>
> - Paper title could better reflect the contents of the paper.
>
>   We regret that our title failed to reflect the contents well. We are open to changing the title should the system allow it.
>
> - Extension to vector output case.
>
>   In a multimodal multitask setup, late and intermediate fusion linear networks learn every task from every modality with a unique timescale. This can result in multiple phases in the vector output case, where the networks exhibit qualitatively similar behaviors. For instance, there will be at most 4 transitional periods when there are 2 input modalities and the output is 2-dimensional; see [image](https://postimg.cc/G4Hq1Dmr).
>
>   We will discuss the vector output case in the final revision.
>
> - Different depth between pathways.
>
>   Thank you for the useful suggestion, we have derived the time ratio for unequal depths and added it to the Appendix. The time ratio is given in Eq 57-58.
>
>   The new factor taken into consideration is that deeper linear networks learn faster compared to shallower linear networks with the same initialization per layer. The reason is that small initialization accumulates through multiplication in deep linear networks, making the gradient update around initialization very small. The qualitative conclusions derived for networks with equal depth between pathways still hold for unequal depth cases. Additionally, a larger difference in the pre-fusion layer depth can prolong the unimodal phase.

---

> ### Author Response · Authors · 2023-11-18
> **Response to Reviewer W8DV (part 2)**
>
> - How do the weights evolve in the pre and post fusion layers?
>
>   We added a section on Feature Evolution in Appendix E and uploaded videos in supplementary material to elaborate on this.
>
>   The weights in linear networks stay rank-one and aligned as described in Eq 39. The first-layer weights in linear networks are the features and have different structures depending on the task. Intermediate layers stay aligned with their consecutive layers and only evolve in scale. Our added Appendix section discusses the specifics of the first-layer weight evolution in early, intermediate, and late fusion linear networks.
>
>   As for the norm of layers, the conservation law given in Eq 36 hold true during training. Assume modality A is learned first. In intermediate fusion networks, the norm of post-fusion layers and pre-fusion layers of modality A grow to $|| \boldsymbol \Sigma_{yx_A} \boldsymbol \Sigma_A ||^{\frac1L}$ during the first transitional period. After a unimodal phase, the norm of pre-fusion layers of modality B grows to $u_B^\infty$ while norm of post-fusion layers adjusts to $u^\infty$ and norm of pre-fusion layers of modality A adjusts to $u_A^\infty$ where
>   $$
>   u_A^\infty = w_A^{\frac1L} \left[ 1+ \left(\frac{w_B}{w_A}\right)^{\frac2{L_f}} \right]^{-\frac{L-L_f}{2L}}
>   $$
>
>   $$
>   u_B^\infty = w_B^{\frac1L} \left[ 1+ \left(\frac{w_A}{w_B}\right)^{\frac2{L_f}} \right]^{-\frac{L-L_f}{2L}}
>   $$
>
>
>   $$
>   u^\infty = \left( w_A^{\frac2{L_f}} + w_B^{\frac2{L_f}} \right)^{\frac{L_f}{2L}}
>   $$
>   We use $w_A, w_B$ to denote the norm of the converged total weights for modality A,B, which are the two blocks in the converged total weights $\boldsymbol \Sigma_{yx} \boldsymbol \Sigma$.
>
> - When the paper says: "In essence, an early fusion point allows the weaker modality to benefit from the stronger modality's learning in the post-fusion layers:" Are there settings where this can be harmful as well, or would the larger scale of the weights always help learning?
>
>   We did not mean to imply that the larger scale of weights always helps learning, only that it speeds up learning in the feature learning regime. We note that this may or may not aid generalization depending on the specifics of the task. The unimodal phase arises because different modalities are learned at different speeds. The weaker modality benefits from the stronger modality's learning in the post-fusion layers because the difference in speeds is brought closer, shortening the unimodal phase. However, the larger scale of weights alone cannot lead to conclusions on the length of the unimodal phase. We will clarify these points in the revision.
>
> - Do the results apply to more complicated covariance matrices?
>
>   Yes! Our results apply to arbitrary covariance matrices. We made no assumptions on the input covariances for our derivations. In the figures, our simulations are done with 2x2 covariance matrices, but they are not always diagonal. We varied the correlation coefficient $\rho$ in Fig 3 and Fig 5b. The covariance matrix is not diagonal for any $\rho \neq 0$. It is an important part of our results that a stronger correlation between input modalities prolongs the unimodal phase.
>
>   The reason for using 2x2 covariance is that metrics for variances inside a modality and correlations across modalities are very clear in 2x2 covariance matrices, which allows us to systematically sweep the variance and correlation parameters in Fig 3 and 5. Furthermore, we have found that covariance matrices with higher dimensions do not give rise to any qualitatively different behavior as long as the input dimension is not larger than the number of training samples.

---

> ### Author Response · Authors · 2023-11-18
> **Response to Reviewer W8DV (part 3: minor clarifications)**
>
> ## Minor Clarifications
>
> - Norm notation
>
>   We added a footnote in our manuscript clarifying that we use $|| \cdot ||$ to denote the L2 norm of a vector or the Frobenius norm of a matrix.
>
> - Product notation
>
>   We added a definition to clarify that we abuse the product notation to represent the ordered product of matrices with the largest index on the left and smallest on the right. For example, $\prod_{i=1}^L \boldsymbol W^i = \boldsymbol W^L \boldsymbol W^{L-1} \cdots \boldsymbol W^1$.
>
> - Experimental details
>
>   We added an Implementation Details section at the end of Appendix. The amount of samples for all experiments is 4096. We randomly draw 4096 input data points from a normal distribution with zero mean and a certain covariance matrix. The dataset is fixed for a single run of training a network. We use the regular gradient descent on full batch (batchsize=4096) with learning rate 0.04.
>
> - Why it is ideal for modality learned first to lead to larger decrease in loss?
>
>   We apologize for the inappropriate wording of "ideal" and have deleted this sentence from the manuscript.
>
>   The comparison we intend to make is between the two cases: (i) the network learns modality A first, goes through a unimodal phase, and learns modality B; (ii) the network learns modality B first, goes through a unimodal phase, and learns modality A. Assume learning modality A yields a larger decrease in loss. Then we consider case (i) to be better than (ii), since one would have a better model if training is terminated during the unimodal phase. However, we acknowledge that both cases are not ideal in the sense that the unimodal phase exists.
>
> - Unclear wording: "a smaller initialization scale exacerbates the impediment to learning modality A compared to modality B, yielding a larger time ratio."
>
>   We have re-phrased this sentence and adopted a more direct wording: "Even amongst cases that all fall into the rich feature learning regime, the initialization scale has an effect on the time ratio, with a larger time ratio for a larger initialization scale."
>
> - Why do the results require a small initialization?
>
>   From the technical side, our derivations utilized the conservation law $\boldsymbol W_1 \boldsymbol W_1^\top = \boldsymbol W_2^\top \boldsymbol W_2$ between layers, which is true for small initialization but often untrue for large initialization. We will clarify in the final revision.
>
>   From the motivation side, the neural tangent kernel community extensively has studied that neural networks with large initialization are in the lazy learning regime and networks with small initialization are in the rich feature learning regime; see [1,2]. In many settings where representation learning is important, neural networks generalize better when trained in the rich rather than lazy regime. Practical neural network systems often learn structured representations. We thus study networks with small initialization to understand this rich feature learning behavior.
>
>   [1] Woodworth, et al. "Kernel and rich regimes in overparametrized models." COLT. 2020.
>
>   [2] Chizat et al. "On lazy training in differentiable programming." NeurIPS. 2019.

---

> > ### Comment · Reviewer_W8DV · 2023-11-22
> >
> > I thank the authors for their detailed reply and updated revision. The authors have addressed my questions and I believe have improved their manuscript. I have updated my score to an accept from a borderline accept to reflect this.
> >
> > As a minor clarification, regarding, *"The new factor taken into consideration is that deeper linear networks learn faster compared to shallower linear networks with the same initialization per layer. The reason is that small initialization accumulates through multiplication in deep linear networks, making the gradient update around initialization very small"*, why would deeper linear networks learn faster with a small initialization?

---

> ### Author Response · Authors · 2023-11-22
> **Thanks and a minor clarification**
>
> Thank you very much for reviewing our rebuttal and updating the acore!
>
> And thanks again for a sharp catch in our response! The quoted sentence was indeed a slip-up. What we intend to write is *"deeper linear networks learn slower compared to shallower linear networks with the same initialization per layer"*. As indicated by Eq 44, the time when the first modality gets learned scale with $u_0^{2-L}$. Since the initialization scale $u_0$ is usually smaller than 1, the time increases with larger depth $L$. Hence, with $u_0$ held fixed, a deeper linear network experiences a longer plateau before its first transitional period. This has been studied by prior literature on regular linear networks. Figure E1(a) in [1] and Figure S1 in [2] both illustrate this point.
>
> [1] Atanasov, et al. "Neural Networks as Kernel Learners: The Silent Alignment Effect." ICLR. 2022.
>
> [2] Saxe, et al. "A mathematical theory of semantic development in deep neural networks." PNAS. 2019.

---

### Author Response · Authors · 2023-11-18
**Rebuttal revision summary**

We thank all the reviewers for their thoughtful questions and suggestions. Bases on their input, we have revised the paper with the following changes.

Added Sections

- Appendix D.4: Time Ratio for Unequal Depth

  Derived the time ratio for intermediate fusion linear networks with unequal depth between modalities.

- Appendix E: Feature Evolution in Multimodal Linear Networks

  Compared early fusion with late fusion from the new feature evolution perspective.

- Appendix G: Underparameterization and Overparameterization in Multimodal Linear Networks

  Brought up the topic of underparameterization and overparameterization to help motivate our studies into the transient unimodal phase which can lead to permanent unimodal bias.

- Appendix I: Implementation Details

Revised Sections

- Added Fig. 7 to show the learning trajectories of late fusion linear network when the input modalities are collinear.
- Revised Appendix H.2 to elaborate on the XOR and linear case.

Added Supplementary Material

- Uploaded videos for Feature Evolution in Multimodal Linear Networks.
- Uploaded videos for Feature Evolution in Two-Layer Early/Late Fusion ReLU Networks.

Minor Changes

- Added notation clarifications and rephrased a few sentences in the main text.
- Fixed a few typos.

---

### Meta-Review · Area_Chair_cx3f · 2023-12-05

**Metareview:**

This paper received contrasting reviews, namely, 8, 5, 5, and 3, reached after rebuttal and consequent discussion.

The paper is recognized to have some merits but also several issues. Motivations underlying this work are not well addressed, novelty aspects also seem not well evident and, most of all, experimental analysis and ablations are not well developed nor suitable to demonstrate the validity of the proposed method. Moreover, the presentation of some parts of the methodology results not fully clear.

The authors provided a careful rebuttal, which succeeded in convincing a reviewer to raise his score, but the remaining ones stuck into their initial ratings below threshold. The main reason remains the simplistic experimental analysis.

For these reasons, this paper is deemed not acceptable for publication in ICLR 2024.

**Justification For Why Not Higher Score:**

Most reviews are below threshold, even overlooking the comments of reviewer T6g2.
This paper seems to have serious problems in the experimental part.

**Justification For Why Not Lower Score:**

N/A

---

### Decision · Program_Chairs · 2024-01-16

Reject